# Human serum triggers antibiotic tolerance in *Staphylococcus aureus*

Elizabeth V. K. Ledger[1], Stéphane Mesnage ⬥ [2] & Andrew M. Edwards ⬥ [1✉]

*Staphylococcus aureus* frequently causes infections that are challenging to treat, leading to high rates of persistent and relapsing infection. Here, to understand how the host environment influences treatment outcomes, we study the impact of human serum on staphylococcal antibiotic susceptibility. We show that serum triggers a high degree of tolerance to the lipopeptide antibiotic daptomycin and several other classes of antibiotic. Serum-induced daptomycin tolerance is due to two independent mechanisms. Firstly, the host defence peptide LL-37 induces tolerance by triggering the staphylococcal GraRS two-component system, leading to increased peptidoglycan accumulation. Secondly, GraRS-independent increases in membrane cardiolipin abundance are required for full tolerance. When both mechanisms are blocked, *S. aureus* incubated in serum is as susceptible to daptomycin as when grown in laboratory media. Our work demonstrates that host factors can significantly modulate antibiotic susceptibility via diverse mechanisms, and combination therapy may provide a way to mitigate this.

---

[1] MRC Centre for Molecular Bacteriology and Infection, Imperial College London, Armstrong Rd, London SW7 2AZ, UK. [2] School of Biosciences, University of Sheffield, Sheffield S10 2TN, UK. ✉email: a.edwards@imperial.ac.uk

   **1**

S. *aureus* is a leading cause of invasive infections, resulting in over 12,000 cases of bacteraemia each year in the UK[1]. The choice of treatment depends on the antibiotic susceptibility of the pathogen, with infections caused by methicillin sensitive *S. aureus* (MSSA) usually treated with front-line β-lactams such as oxacillin. However, there are few effective therapeutic options for invasive infections caused by methicillin resistant *S. aureus* (MRSA) strains, with the recommended treatments including vancomycin and daptomycin[2].

Daptomycin, a cyclic lipopeptide antibiotic approved to treat MRSA bacteraemia in 2003, targets membrane phosphatidylglycerol (PG) as well as bactoprenyl-coupled cell wall precursors such as lipid II in a calcium-dependent manner[3]. In addition, daptomycin disrupts the localisation of cell wall biosynthetic machinery, including MurG, further compromising cell wall biosynthesis[4,5]. At supra-inhibitory concentrations, daptomycin also causes membrane disruption, although this does not appear to be necessary for antibacterial activity[4,5].

In vitro, daptomycin is highly potent and rapidly bactericidal but this potency is not observed in vivo, where despite giving high doses intravenously, it can take several days to sterilise the bloodstream[6,7]. This slow rate of clearance allows *S. aureus* to disseminate around the body, leading to the development of secondary infections[8] such as infective endocarditis, osteomyelitis, septic arthritis, meningitis and tissue abscesses[9]. As a result, patients require long hospital stays, leading to high treatment costs, and suffer from high mortality rates[10,11]. Specifically, daptomycin fails to cure 20–30% cases of MRSA bacteraemia and is associated with a 10–20% mortality rate[6,7,12,13]. Therefore, it is important to understand why treatment failure occurs so that new approaches can be developed to improve patient survival rates.

Although daptomycin resistance (typically referred to as non-susceptibility) can occur during therapy, it is not the only explanation for treatment failure and in many patients failure occurs despite the infecting strain being classed as susceptible by antimicrobial susceptibility testing[14,15]. In these cases, a possible explanation for treatment failure is antibiotic tolerance, where a bacterial population survives exposure to a normally lethal concentration of an antibiotic without an increase in minimum inhibitory concentration (MIC)[16]. This tolerance can be due to mutations[17–19] or phenotypic adaptation, a transient phenotype which is thought to be induced by the environment conditions, especially those found in the host[14,15] As the population returns to a susceptible state when it is removed from the host environment, this tolerance is difficult to detect in vitro but may play an important role clinically[20]. Furthermore, there is growing evidence that tolerance is a precursor to the acquisition of antibiotic resistance[21].

The mechanisms responsible for antibiotic tolerance are poorly understood, although it is frequently ascribed to slow rates of bacterial growth and/or low metabolic activity[14,22]. These are thought to compromise the activity of antibiotics that require bacterial division for full activity, such as β-lactams and quinolones. However, as daptomycin is effective against non-growing bacteria[23], this is unlikely to be the basis of the reduced efficacy of daptomycin in vivo and the mechanisms of tolerance towards daptomycin remain unknown.

Although the causes of antibiotic tolerance are not fully understood, there is growing evidence that it can be transiently induced by the host environment during infection. For example, the numerous stresses found in the host, such as nutrient limitation, low pH, pyrexia, oxidative and/or nitrosative stress, antimicrobial peptides and proteases, mean that bacteria are rarely in a rapidly replicating state[14,24,25]. These stresses may restrict bacterial growth directly, for example hypoxia leading to a switch to anaerobic metabolism, which produces less energy than the TCA cycle, or indirectly, for example nutrient limitation activating the stringent response[26–28]. This response results in downregulation of expression of genes required for growth, leading to a reduction in protein synthesis and a slower bacterial growth rate[26]. This has been linked to the development of tolerance towards a range of antimicrobials including penicillin, vancomycin and ciprofloxacin[29].

In addition to restricted bacterial growth rates due to hostile conditions, activation of other bacterial stress responses by host factors has also been implicated in antibiotic tolerance[30,31] and the formation of multidrug tolerant states, including persister cells and small colony variants[32–34]. However, it is not clear whether the mechanisms by which induction of stress responses lead to drug tolerant states is simply due to a reduction in growth or metabolic activity or whether there are other mechanisms involved.

Unfortunately, many of these host-associated stresses are typically not replicated in laboratory culture media, meaning that antibiotic tolerance is difficult to detect and is often missed. Laboratory media lack host-derived nutrients and macromolecules, resulting in significant differences between the metabolism and physiology of *S. aureus* in vitro and in vivo. Additionally, phenotypic studies have identified that growth in model host environments affects properties of the staphylococcal cell envelope, including affecting the structure of the cell wall and the composition and properties of the cell membrane[35–37]. For example, incorporation of serum unsaturated fatty acids affected membrane fluidity, a factor known to influence the susceptibility of *S. aureus* to membrane-targeting antimicrobials[36,38]. However, the impact of such changes on antibiotic susceptibility have not been tested.

Therefore, although there is an increasing awareness that the host environment affects many aspects of *S. aureus* physiology and metabolism, the impact this has on antibiotic susceptibility and the mechanisms involved are poorly understood.

Here, we used human serum as a model host environment of bacteraemia, leading to the discovery of two mechanisms by which host defences induce daptomycin tolerance and the identification of a possible combination therapeutic approach to enhance antibiotic efficacy.

## Results

**Incubation of *S. aureus* in serum results in daptomycin tolerance.** Since daptomycin is frequently used to treat *S. aureus* bloodstream infections[1,39], we used normal human serum as an ex vivo model to examine how the host environment affected daptomycin susceptibility. In this model, we performed antibiotic bactericidal activity assays on *S. aureus* in two states: (i) grown to mid-exponential phase in tryptic soy broth (TSB) to represent in vitro conditions ("TSB-grown") and (ii) incubated for 16 h in human serum to mimic host conditions and provide a reasonable duration between infection onset and treatment initiation ("serum-adapted"; Fig. 1a).

We used a high inoculum of bacteria ($2 \times 10^8$ CFU ml$^{-1}$) for two key reasons. Firstly, daptomycin is licenced for use for treatment of infective endocarditis, which is associated with high tissue densities[40–42]. Secondly, since daptomycin is potently bactericidal against broth-grown bacteria, it was important to have a large dynamic range to be able to detect and characterise changes in tolerance[23].

In agreement with previous reports[43], colony forming unit (CFU) counts of *S. aureus* did not change during the 16 h incubation in serum (Supplementary Fig. 1). Therefore, equal numbers of *S. aureus* CFU from TSB-grown and serum-adapted

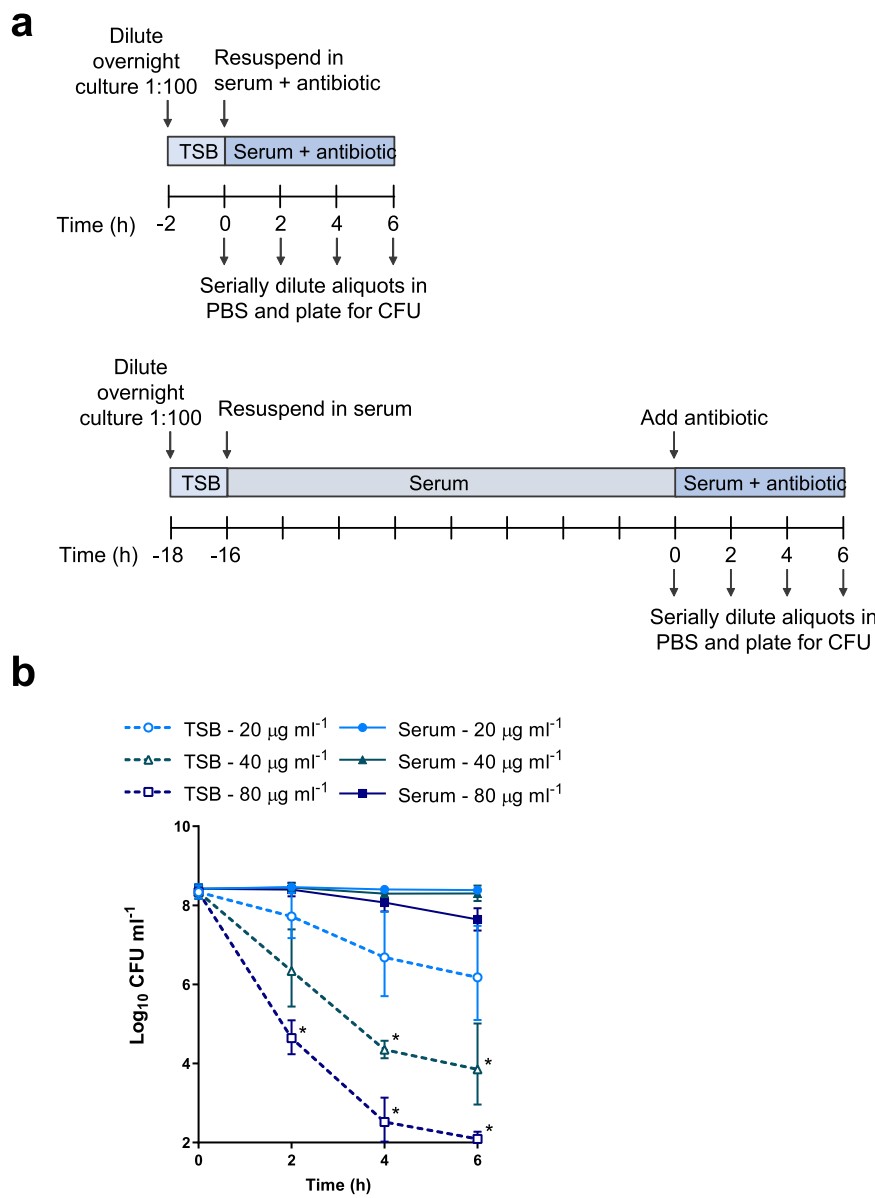

**Fig. 1 Incubation of *S. aureus* in serum results in tolerance towards daptomycin.** Schematic outlining the protocol used to investigate the susceptibility of TSB-grown and serum-adapted cultures of *S. aureus* to daptomycin (**a**). $Log_{10}$ CFU $ml^{-1}$ of TSB-grown and serum-adapted cultures of *S. aureus* USA300 WT over 6 h incubation in serum with 20–80 µg $ml^{-1}$ daptomycin (**b**). Graph represents the geometric mean ± geometric standard deviation of three independent experiments (* for 40 µg $ml^{-1}$ $P = 0.0014$ (4 hr), 0.0472 (6 h) and for 80 µg $ml^{-1}$ $P = 0.0068$ (2 h), 0.0042 (4 h), 0.0003 (6 h) determined by two-way ANOVA with Tukey's *post-hoc* test, TSB-grown compared to serum-adapted at each time-point and dose). TSB, tryptic soy broth; PBS, phosphate-buffered saline; CFU, colony-forming units. Source data are provided as a Source Data file.

cultures were exposed to daptomycin. To control for any effects that serum had directly on the antibiotic, for example via protein binding, bacterial survival assays of both TSB-grown and serum-adapted cultures were performed in serum (Fig. 1a).

When TSB-grown *S. aureus* were challenged with a range of clinically-achievable concentrations of daptomycin (up to 80 µg $ml^{-1}$; equivalent to peak serum concentrations observed in patients[44]), dose- and time-dependent killing over 6 h was observed (Fig. 1b). There were lower levels of bacterial survival at higher daptomycin concentrations and later time points, confirming that TSB-grown *S. aureus* were susceptible to daptomycin and that despite serum protein binding, high levels of killing were achieved in serum (Fig. 1b). By contrast, when serum-adapted *S. aureus* were exposed to daptomycin, killing was completely abolished at both 20 and 40 µg $ml^{-1}$ daptomycin, while 17% of the initial inoculum survived exposure

to 80 µg $ml^{-1}$ daptomycin (Fig. 1b). Therefore, at 80 µg $ml^{-1}$ daptomycin, serum adaptation led to a >200,000-fold increase in bacterial survival compared to TSB-grown *S. aureus*.

To test whether daptomycin tolerance was simply a consequence of the lack of bacterial replication or the initiation of a starvation stress response due to the 16 h incubation period, we repeated the assay with bacteria incubated in serum, TSB or PBS for 16 h. In contrast to bacteria incubated in serum, and in keeping with previous work[23], *S. aureus* incubated in TSB or PBS remained highly susceptible to killing by daptomycin, indicating that serum specifically triggered a high level of daptomycin tolerance in *S. aureus* (Supplementary Fig. 2).

Next, we tested whether this serum-induced tolerance was unique to daptomycin or whether it also occurred with other classes of antimicrobial. Serum adaptation conferred tolerance

towards two representative antimicrobial peptides (AMPs), nisin and gramicidin, demonstrating that serum adaptation was able to protect *S. aureus* not only from daptomycin but also from other membrane-targeting antimicrobials (Supplementary Fig. 3a, b). Moreover, serum adaptation resulted in high levels of tolerance towards three additional bactericidal antibiotics with diverse mechanisms of action, vancomycin (cell wall synthesis inhibition), nitrofurantoin (DNA damage) and gentamicin (protein synthesis inhibition) (Supplementary Fig. 3c–e).

Taken together, adaptation to the host environment conferred high levels of tolerance towards the last-resort antibiotic daptomycin. Serum also induced tolerance towards other membrane-targeting antimicrobials with a similar mechanism of action to daptomycin and towards other diverse antibiotics with various modes of killing.

**Daptomycin does not bind to *S. aureus* incubated in serum.** Despite potent activity in vitro, daptomycin is associated with high rates of relapse and mortality, possibly due to the tolerance phenotype described above[6,7,12,13]. We therefore decided to determine the mechanism by which adaptation to serum conferred daptomycin tolerance and thereby identify opportunities to enhance treatment efficacy.

We first determined whether incubation in serum prevented daptomycin from interacting with the bacterial membrane. To do this, TSB-grown and serum-adapted cells were exposed to the fluorescent BoDipy-labelled daptomycin in serum, and aliquots taken at each time point to measure the levels of cell-associated antibiotic. A large increase in the fluorescence of TSB-grown cultures was observed during the first 2 h, followed by a gradual increase over the following 4 h (Fig. 2a). By contrast, a smaller increase in fluorescence during the first 2 h was observed with serum-adapted *S. aureus*, with no further increase after this time point, indicating reduced binding of daptomycin to serum-adapted bacteria compared to TSB-grown cells (Fig. 2a). Aliquots taken at the 2 h time point were fixed, co-stained with the lipophilic dye Nile Red to visualise cell membranes and analysed by fluorescence microscopy. In agreement with Fig. 2a, this demonstrated that *S. aureus* were bound by higher levels of BoDipy-daptomycin when in a TSB-grown state than in a serum-adapted state (Fig. 2b).

Since the clinically relevant concentration of daptomycin used here is $160 \times$ MIC in TSB, it would be expected to cause membrane depolarisation and permeabilisation[4,5]. Therefore, we next determined whether serum adaptation protected *S. aureus* from this damage[45]. To do this, we measured membrane potential using the voltage-sensitive fluorescent dye $DiSC_3(5)$[46] and the ability of *S. aureus* to exclude propidium iodide (PI) as an indication of membrane permeability[45]. TSB-grown and serum-adapted cultures of *S. aureus* were exposed, or not, to $80 \, \mu g \, ml^{-1}$ daptomycin for 6 h in human serum and then $DiSC_3(5)$ or PI was added and the fluorescence measured. In each case, exposure of TSB-grown cells to $80 \, \mu g \, ml^{-1}$ daptomycin led to a significant increase in fluorescence compared to untreated cells, demonstrating that daptomycin resulted in membrane depolarisation and permeabilisation (Fig. 2c, d). By contrast, there was no significant increase in the fluorescence of serum-adapted bacteria after exposure to daptomycin, showing that serum adaptation prevented daptomycin-induced membrane damage (Fig. 2c, d).

Taken together, these results indicated that high levels of daptomycin bound to TSB-grown *S. aureus*, resulting in membrane permeabilisation, depolarisation and rapid cell death. By contrast, incubation of *S. aureus* in serum resulted in daptomycin tolerance by significantly reducing the amount of daptomycin that bound to the membrane and thereby greatly reducing the associated membrane damage and cell death.

**LL-37 in serum triggers daptomycin tolerance via activation of GraRS.** The next aim was to identify the specific factor(s) in serum responsible for triggering daptomycin tolerance and the staphylococcal system(s) activated. Since two-component systems (TCS) are a key mechanism for sensing many environmental signals, we carried out a screen of mutants defective for each non-essential TCS in the *S. aureus* USA300 genome to determine whether any were required for serum-induced tolerance[47]. To do this, serum-adapted cultures of *S. aureus* JE2 WT and transposon mutants defective for each sensor kinase were exposed to $80 \, \mu g \, ml^{-1}$ daptomycin for 6 h and survival measured by CFU counts. All mutants survived as well in serum as the WT strain (Supplementary Fig. 4), and as expected, serum adaptation conferred daptomycin tolerance on the WT strain (Fig. 3a). By contrast, mutants defective for VraS or GraS showed ~100-fold lower levels of survival compared to WT (Fig. 3a). Complementation of the *graS*::Tn and *vraS*::Tn mutant strains with *graXRS* or *vraUTSR*, respectively, expressed from their native promoters on a low copy number plasmid completely restored daptomycin tolerance to WT levels (Supplementary Fig. 5a, b).

Next, we determined whether either of the VraSR or GraRS signalling systems were activated by serum. To do this, fluorescent reporters were constructed, where the expression of *gfp* was placed under the control of a promoter of a gene regulated by either TCS. To measure induction of VraSR and GraRS signalling, the *vraX* and *dltA* promoters were used as these are known to be regulated by VraSR and GraRS, respectively[48–50]. Strains containing these reporter constructs were exposed to human serum and fluorescence measured over time. An increase in *vraX* expression was not detected on exposure to serum in either the WT or *vraS*::Tn mutant background, providing no evidence that serum induced signalling of this system (Fig. 3b). By contrast, when the WT strain containing the GraRS reporter was exposed to serum there was a rapid increase in GFP fluorescence over the first 3 h, but no change in the fluorescence of the *graS*::Tn mutant reporter strain (Fig. 3c), indicating that serum triggered GraRS signalling in *S. aureus*.

To investigate whether activation of GraRS was capable of inducing daptomycin tolerance, we induced signalling by incubation of *S. aureus* in RPMI 1640 supplemented with sub-inhibitory concentrations of colistin, a known trigger of GraRS[50] (Supplementary Fig. 6a). RPMI 1640 was chosen as it is a host-mimicking cell culture medium that lacks the factor present in serum that activates GraRS. Exposure of the P*dltA*-*gfp* reporter strains to colistin led to dose-dependent production of GFP in the WT strain but not in the *graS*::Tn mutant (Supplementary Fig. 6a), confirming that colistin was triggering GraRS. Next, *S. aureus* was incubated for 16 h in RPMI 1640 supplemented, or not, with colistin and then exposed to daptomycin. Incubation in media alone did not confer daptomycin tolerance, with just 0.1% of the initial inoculum surviving exposure to the antibiotic (Supplementary Fig. 6b). Supplementation of RPMI 1640 with colistin led to dose-dependent increases in bacterial survival in the WT strain, with the highest concentration of colistin triggering a 70-fold increase in survival of the inoculum, relative to bacteria not exposed to colistin (Supplementary Fig. 6b). By contrast, no significant increase in survival was seen in the *graS*::Tn mutant when incubated in RPMI 1640 containing colistin, compared to media alone (Supplementary Fig. 6b).

Next, we investigated whether GraRS signalling could be inhibited pharmacologically to block the development of daptomycin tolerance. Verteporfin is a licenced photosensitising drug used to treat macular degeneration which has been reported to also inhibit GraS[51]. Bacterial adaptation was carried out in serum, or serum supplemented with sub-lethal concentrations of verteporfin, before cultures were exposed to daptomycin. As sub-

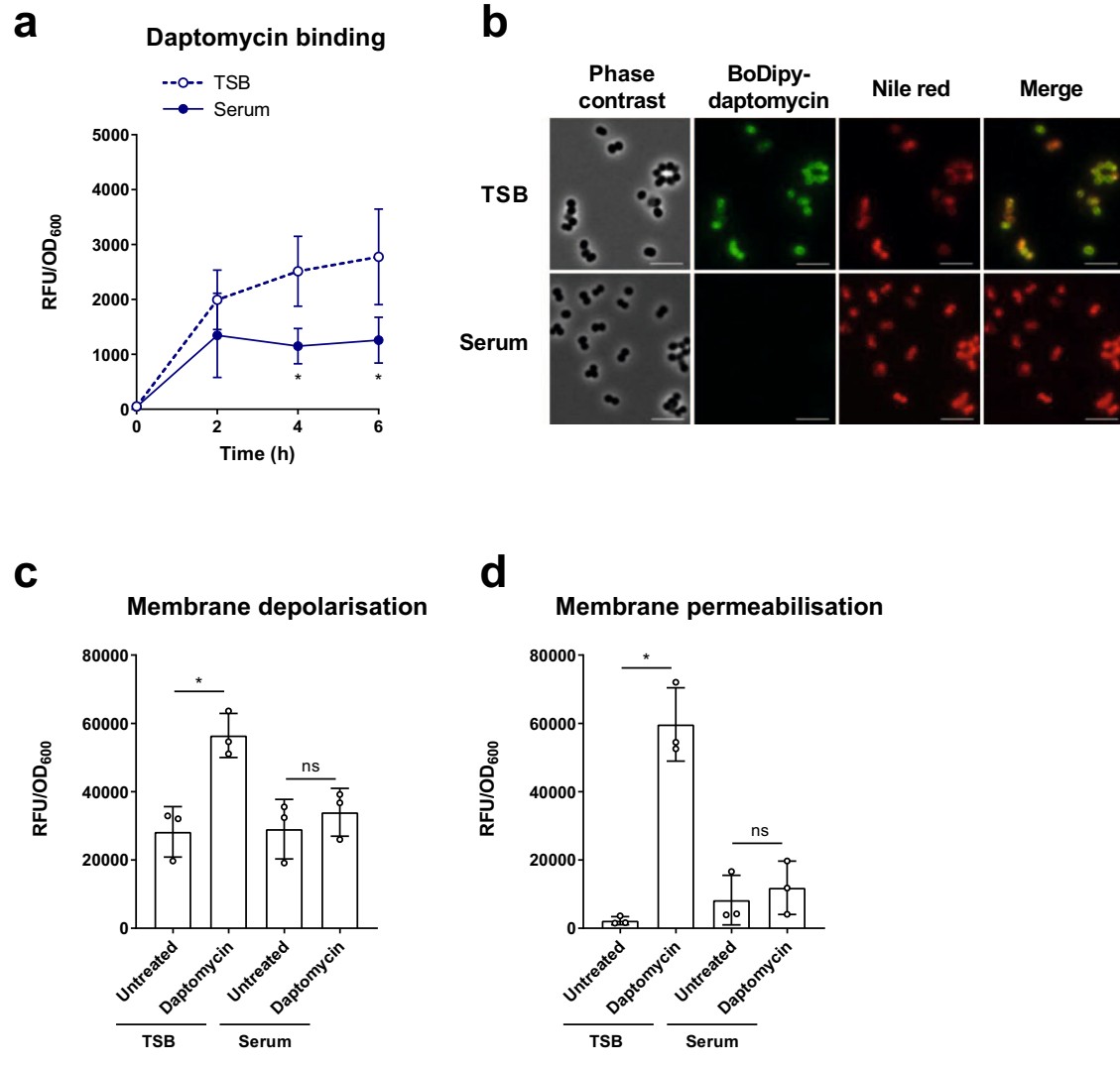

**Fig. 2 Daptomycin does not bind to or disrupt the membranes of *S. aureus* incubated in serum.** Cell-associated fluorescence of TSB-grown and serum-adapted cultures of *S. aureus* USA300 WT over a 6 h incubation with 320 µg ml$^{-1}$ BoDipy-daptomycin (**a**). Cells after 2 h daptomycin exposure from panel (**a**) were fixed and analysed by phase contrast and fluorescence microscopy (**b**). Cells were co-stained with 10 µg ml$^{-1}$ nile red to visualise cell membranes. Scale bars, 5 µm. DiSC$_3$(5) (**c**) and propidium iodide (**d**) fluorescence of TSB-grown and serum-adapted cultures after 6 h exposure, or not, to 80 µg ml$^{-1}$ daptomycin in serum. Fluorescence values were divided by OD$_{600}$ measurements to normalise for changes in cell density which occurred throughout the assays. Graphs represent the mean ± standard deviation of three independent experiments (NS = not significant ($P > 0.05$). *For (**a**), $P = 0.0276$ (4 h), 0.0132 (6 h). For (**c**), $P = 0.0033$ and (**d**), $P < 0.0001$. Data in (**a**) were analysed by two-way ANOVA with Sidak's *post-hoc* test (TSB-grown vs serum-adapted at each time-point). Data in (**c**) and (**d**) were analysed by two-way ANOVA with Tukey's *post-hoc* test (untreated vs daptomycin-exposed). TSB, tryptic soy broth; RFU, relative fluorescence units; OD$_{600}$, optical density at 600 nm. Source data are provided as a Source Data file.

lethal concentrations of verteporfin were used, no decreases in CFU count were observed during this incubation (Supplementary Fig. 7). Consistent with a role for GraRS in daptomycin tolerance, incubation of *S. aureus* in serum in the presence of verteporfin led to a dose-dependent decrease in tolerance, which resulted in 100-fold increase in bacterial killing when 80 µg ml$^{-1}$ verteporfin was used (Fig. 3d). By contrast, supplementation of serum with verteporfin had no significant effect on the tolerance of the *graS*::Tn mutant to daptomycin (Fig. 3d).

Next, we aimed to identify the factor in serum responsible for activating GraRS signalling and thereby daptomycin tolerance. To do this, we first fractionated serum using a series of different molecular weight dialysis membranes. We used a 0.5 kDa cut-off to remove ions and small molecules such as amino acids and sugars, a 5 kDa cut-off to additionally remove peptides, and a 100 kDa cut-off to also remove larger proteins such as serum albumin. We then assessed whether the dialysed serum retained

the ability to trigger daptomycin tolerance. We found that the serum factor responsible for triggering tolerance was between 0.5 and 5 kDa (Supplementary Fig. 8a). To confirm this, we demonstrated that the <5 kDa fraction was sufficient to trigger tolerance (Supplementary Fig. 8b).

Since GraRS is activated by certain AMPs[52], which are often ~5 kDa in size, we tested a range of peptides present in serum to identify the trigger of tolerance. To do this, the P*dltA-gfp* reporter strains were exposed to sub-inhibitory concentrations of AMPs and the ability of each AMP to trigger GraRS activation was determined. LL-37 led to dose-dependent induction of GraRS in the WT background (Fig. 3e). For example, at 8 h, 20, 40, and 80 µg ml$^{-1}$ LL-37 led to 1.6-, 2.7- and 3.6-fold increases in *dltA* expression respectively compared to samples not exposed to LL-37 (Fig. 3e). By contrast, no induction was observed in the *graS*::Tn mutant (Fig. 3e). Human neutrophil peptide-1 (hNP-1) and platelet AMPs did not induce GraRS signalling, while

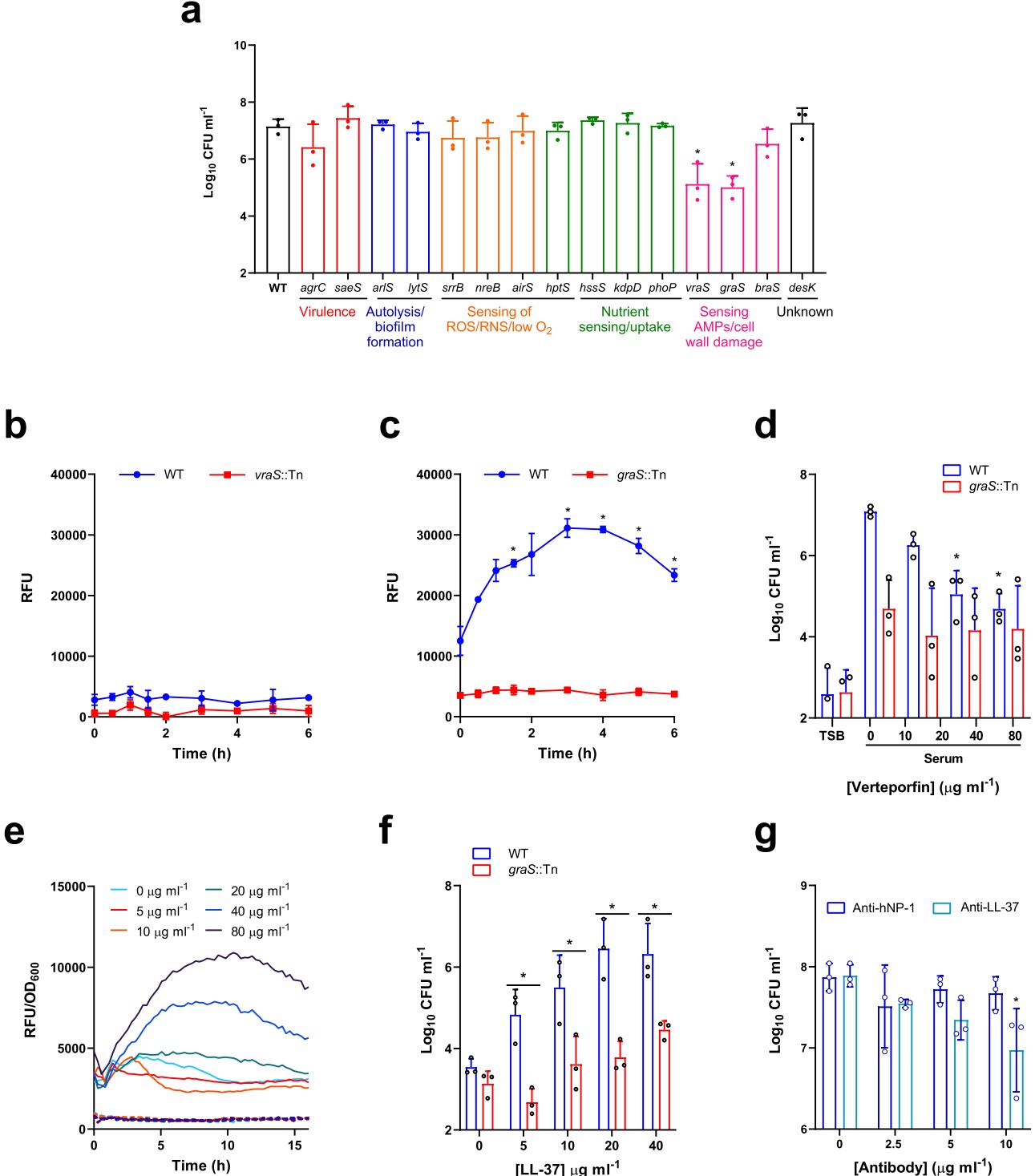

dermcidin only induced signalling very weakly (Supplementary Fig. 9a–c).

To determine whether LL-37 induced daptomycin tolerance, JE2 WT and *graS*::Tn strains were incubated for 16 h in RPMI 1640 alone, or supplemented with various sub-inhibitory concentrations of LL-37, before addition of daptomycin. As it was used at a sub-inhibitory concentration, LL-37 had no effect on bacterial survival (Supplementary Fig. 10). Incubation in media alone did not confer daptomycin tolerance, but as the concentration of LL-37 increased, the daptomycin tolerance of the WT strain also increased, with up to 200-fold increased

survival of bacteria incubated with the highest LL-37 concentration compared with medium without peptide (Fig. 3f). By contrast, no increase in survival of the *graS*::Tn mutant was observed on addition of LL-37 (Fig. 3f). In agreement with the reporter data, neither hNP-1, dermcidin nor platelet AMPs triggered daptomycin tolerance (Supplementary Fig. 9d–f).

To confirm that native serum LL-37 was responsible for triggering tolerance in serum, a sheep polyclonal anti-LL-37 IgG antibody was assessed for its ability to prevent the induction of daptomycin tolerance in *S. aureus*. As a negative control, serum was pre-incubated with a sheep IgG targeting hNP-1, a peptide

**Fig. 3 LL-37 in serum triggers daptomycin tolerance through activation of the GraRS two-component system.** Log$_{10}$ CFU ml$^{-1}$ of serum-adapted *S. aureus* JE2 WT and transposon mutants defective for the sensor components of various TCS after 6 h incubation with 80 μg ml$^{-1}$ daptomycin (**a**). GFP fluorescence over a 6 h exposure of TSB-grown *S. aureus* JE2 WT and the *vraS*::Tn mutant containing P*vraX-gfp* (**b**) or JE2 WT and the *graS*::Tn mutant containing P*dltA-gfp* (**c**) to human serum. Log$_{10}$ CFU ml$^{-1}$ after 6 h exposure to 80 μg ml$^{-1}$ daptomycin of *S. aureus* JE2 WT and the *graS*::Tn mutant which had been TSB-grown or pre-incubated for 16 h in serum supplemented with indicated concentrations of verteporfin (**d**). TSB-grown *S. aureus* JE2 WT (solid lines) and the *graS*::Tn mutant (dashed lines) containing P*dltA-gfp* were exposed to various concentrations of LL-37 (5–80 μg ml$^{-1}$) in RPMI 1640 and GFP fluorescence (RFU) and OD$_{600}$ were measured every 15 min for 16 h (**e**). Fluorescence values were divided by OD$_{600}$ measurements to normalise for changes in cell density. Log$_{10}$ CFU ml$^{-1}$ of WT and *graS*::Tn mutant strains which had been incubated for 16 h in RPMI 1640 supplemented with indicated concentrations of LL-37 and then exposed to 80 μg ml$^{-1}$ daptomycin for 6 h (**f**). Log$_{10}$ CFU ml$^{-1}$ of *S. aureus* JE2 WT incubated for 16 h in serum supplemented with indicated concentrations of antibody targeting hNP-1 or LL-37 and then exposed to 80 μg ml$^{-1}$ daptomycin for 6 h (**g**). Graphs in (**a**), (**d**), (**f**) and (**g**) represent the geometric mean ± geometric standard deviation of three independent experiments. Graphs in (**b**), (**c**) and (**e**) represent the mean ± standard deviation of three independent experiments except panel (**e**) where error bars have been omitted for clarity. Data in (**a**) were analysed by one-way ANOVA with Dunnett's *post-hoc* test (*$P < 0.0001$ (*vraS*), $<0.0001$ (*graS*) vs WT). Data in (**b**) and (**c**) were analysed by two-way ANOVA with Dunnett's *post-hoc* test (* for (**c**), $P = 0.0416$ (90 min), 0.0227 (3 h), 0.0491 (4 h), 0.0248 (5 h), 0.0422 (6 h)). Data in (**d**) and (**g**) were analysed by two-way ANOVA with Dunnett's *post-hoc* test (* for (**d**), $P = 0.01$ (40 μg ml$^{-1}$), 0.0024 (80 μg ml$^{-1}$); * for (**g**), $P = 0.0044$; serum + verteporfin/antibody vs serum alone). Data in (**f**) were analysed by two-way ANOVA with Sidak's *post-hoc* test (*$P = 0.0006$ (5 μg ml$^{-1}$), 0.0023 (10 μg ml$^{-1}$), $<0.0001$ (20 μg ml$^{-1}$), 0.0026 (40 μg ml$^{-1}$). WT vs *graS*::Tn). RFU, relative fluorescence units; CFU, colony-forming units, OD$_{600}$, optical density at 600 nm; hNP-1, human neutrophil peptide 1. Source data are provided as a Source Data file.

which did not activate GraRS signalling or trigger daptomycin tolerance (Supplementary Fig. 9a, d). As expected, the antibody did not affect bacterial survival in serum (Supplementary Fig. 10). In the absence of antibody pre-incubation, high levels of daptomycin tolerance were observed, however, as the concentration of anti-LL-37 antibody increased, the ability of the serum to induce daptomycin tolerance decreased in a dose-dependent manner (Fig. 3g). By contrast, pre-incubation with the anti-hNP-1 antibody had no effect on daptomycin tolerance at any of the concentrations used (Fig. 3g). Taken together, these data demonstrate that serum triggered daptomycin tolerance via LL-37 mediated activation of GraRS signalling.

**GraRS-mediated changes in surface charge do not explain tolerance.** The next aim was to determine how activation of GraRS by serum conferred daptomycin tolerance. GraRS regulates many genes, including those encoding the lysyl-phosphatidylglycerol (LPG) synthase and flippase, MprF, and the DltABCD system, which modifies teichoic acids with D-alanine[52–54]. Upregulation of these genes results in an increase in positive surface charge, which has been proposed to reduce daptomycin susceptibility through charge-mediated repulsion[55].

Therefore, we measured the surface charge of TSB-grown and serum-adapted cultures of *S. aureus* using a fluorescently labelled cationic molecule, fluorescein isothiocyanate-poly-L-lysine (FITC-PLL). TSB-grown and serum-adapted cultures were incubated with FITC-PLL, washed to remove unbound dye, and fixed. Analysis of the fluorescence of the surface of individual cells demonstrated that serum-adapted bacteria bound significantly less FITC-PLL, and so were more positively charged, than TSB-grown cells (Fig. 4a).

Next, we aimed to determine whether either of the main systems involved in modulating surface charge in *S. aureus*, MprF or DltABCD, were required for tolerance. To do this, TSB-grown and serum-adapted cultures of mutants in each system (*mprF*::Tn and Δ*dltD*, respectively) were exposed to 80 μg ml$^{-1}$ daptomycin and survival compared to WT strains. Neither mutant was more susceptible to daptomycin than WT when in a TSB-grown state (Fig. 4b, c). Serum adaptation conferred daptomycin tolerance on the *mprF*::Tn mutant, with survival of 100% of the mutant population observed at 6 h (Fig. 4b). By contrast, serum adaptation did not protect the Δ*dltD* mutant from daptomycin, with a 1000-fold reduction in CFU counts after 6 h exposure to the antibiotic (Fig. 4c). This defect in daptomycin tolerance was fully complemented by the expression of *dltD* on a plasmid from

its native promoter (Supplementary Fig. 12). Therefore, teichoic acid D-alanylation, but not LPG synthesis, was required for daptomycin tolerance in serum.

To explore the requirement for teichoic acid D-alanylation for tolerance further, we next determined whether incubation in serum affected the amount of wall teichoic acid (WTA) present or the degree to which it was modified. To do this, WTA was extracted from TSB-grown and serum-adapted cultures of *S. aureus*, analysed by native polyacrylamide gel electrophoresis (PAGE), visualised by alcian blue staining, and quantified. This demonstrated two-fold higher levels of WTA in serum-adapted than TSB-grown bacteria (Fig. 4d). The levels of D-alanine present in WTA extracts were quantified using an enzyme-based spectrophotometric assay, demonstrating a three-fold increase in the D-alanine content of serum-adapted cells compared to TSB-grown (Fig. 4e). Finally, to investigate whether this increase in D-alanylated WTA was mediating tolerance, TSB-grown cultures were incubated for 16 h in serum supplemented, or not, with one of three WTA synthesis inhibitors, before addition of daptomycin and measurement of survival. Supplementation of serum with either tarocin A1 or tunicamycin completely prevented the increase in D-alanylated WTA which occurred during serum adaptation (Fig. 4d, e). However, this inhibition did not affect serum-mediated daptomycin tolerance (Fig. 4f), demonstrating that despite D-alanylated WTA being required for tolerance, the increase in this modified polymer observed in serum was not responsible for tolerance.

**GraRS-mediated changes to the cell wall partially explain tolerance.** Having ruled out the MprF-mediated synthesis of LPG and the DltABCD-mediated increase in D-alanylated WTA as direct causes of serum-induced daptomycin tolerance, we next investigated whether other GraRS-regulated genes mediated tolerance. As many genes involved in cell wall metabolism, including several peptidoglycan hydrolytic enzymes[53,56], are regulated by GraRS we tested whether there were differences in the peptidoglycan layer between TSB-grown and serum-adapted *S. aureus*.

Cell walls were extracted from TSB-grown and serum-adapted bacteria, the WTA removed by acid hydrolysis and the resulting peptidoglycan freeze-dried and its mass determined. This demonstrated that serum-adapted *S. aureus* contained approximately 5-fold more peptidoglycan than TSB-grown cells (Fig. 5a). This accumulation of peptidoglycan during incubation in serum was partially dependent on GraRS as serum-adaptation led to a smaller increase in the peptidoglycan content of the *graS*::Tn

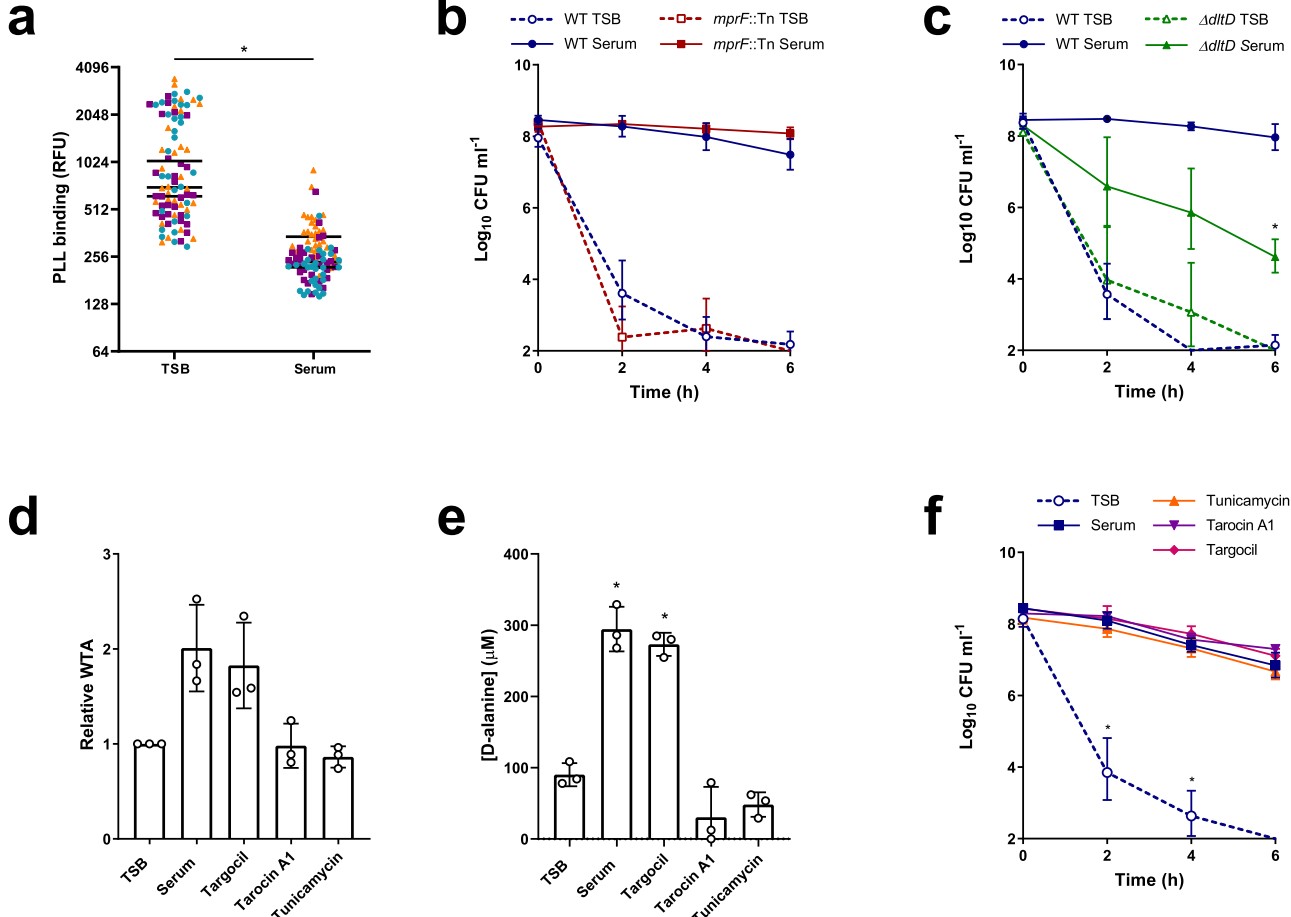

**Fig. 4 GraRS-mediated changes in surface charge do not explain serum-induced tolerance.** FITC-PLL binding to TSB-grown and serum-adapted cultures *S. aureus* JE2 WT. Cultures were incubated with 80 µg ml$^{-1}$ FITC-PLL, washed, fixed and analysed by fluorescence microscopy (**a**). The fluorescence of 30 cells per biological replicate (90 cells total per condition) was measured and the mean of each replicate is indicated. Colours represent different biological replicates. Log$_{10}$ CFU ml$^{-1}$ of TSB-grown and serum-adapted cultures of *S. aureus* JE2 WT and the *mprF*::Tn mutant (**b**) and the *ΔdltD* mutant (**c**) during a 6 h exposure to 80 µg ml$^{-1}$ daptomycin in serum (**b**). WTA extracts from TSB-grown bacteria and cultures incubated in serum supplemented, or not, with 128 µg ml$^{-1}$ targocil, 64 µg ml$^{-1}$, tarocin A1, or 128 µg ml$^{-1}$ tunicamycin for 16 h were analysed by native PAGE with alcian blue staining and quantified using ImageJ (**d**). Concentrations of D-alanine in WTA extracts from panel (**d**) were determined spectrophotometrically using an enzyme-based assay and by interpolating values from a standard curve generated from known D-alanine concentrations (**e**). Log$_{10}$ CFU ml$^{-1}$ over 6 h exposure to 80 µg ml$^{-1}$ daptomycin of TSB-grown *S. aureus* or cultures which had been incubated in serum supplemented, or not, with 128 µg ml$^{-1}$ targocil, 64 µg ml$^{-1}$ tarocin A1 or 128 µg ml$^{-1}$ tunicamycin for 16 h (**f**). Data in (**a**) were analysed by a Mann–Whitney test (*$P < 0.0001$). Data in (**b**) and (**c**) represent the geometric mean ± geometric standard deviation of three independent experiments and were analysed by two-way ANOVA with Sidak's *post-hoc* test (* for (**c**), $P = 0.0036$ serum-adapted WT vs serum-adapted mutant at 6 h time-point). Data in (**d**) and (**e**) represent the mean ± standard deviation of three independent experiments and were analysed by one-way ANOVA with Dunnett's *post-hoc* test (*$P < 0.0001$ (serum), $<0.0001$ (Targocil) vs TSB-grown). Data in (**f**) were analysed by two-way ANOVA with Dunnett's *post-hoc* test (*$P = 0.0222$ (2 h), 0.0072 (4 h), serum alone vs TSB or serum + WTA synthesis inhibitor). Several points fell below the limit of detection of 100 CFU ml$^{-1}$. TSB, tryptic soy broth; FITC-PLL, fluorescein isothiocyanate-poly-L-lysine; RFU, relative fluorescence units; CFU, colony-forming units; WTA, wall teichoic acid. Source data are provided as a Source Data file.

mutant strain (~2-fold) (Fig. 5a). To corroborate these findings, a fluorescent approach was developed to evaluate the peptidoglycan content by measuring the incorporation of a fluorescent D-amino acid analogue, HADA[57]. This analogue is incorporated into the pentapeptide stem of peptidoglycan by *S. aureus* in the place of D-alanine[57] and so provides a measure of the amount of peptidoglycan synthesised during the incubation with HADA. TSB-grown and serum-adapted *S. aureus* were generated in the presence of HADA, fixed, analysed by fluorescence microscopy, and quantified (Fig. 5B). In agreement with the data in Fig. 5a, this revealed that serum-adapted WT *S. aureus* showed >5-fold higher levels of fluorescence than TSB-grown bacteria, and this increase in HADA fluorescence was reduced in the absence of *graS* (Fig. 5b). Furthermore, similar results were seen with the

*ΔdltD* mutant, which had significantly lower levels of peptidoglycan compared WT cells after incubation in serum, potentially explaining why D-alanylation of WTA was needed for serum-induced daptomycin tolerance (Supplementary Fig. 13).

We next investigated whether incubation in serum resulted in peptidoglycan modifications mediated by the GraRS TCS. Peptidoglycan was purified from WT and *graS*::Tn cells grown in TSB or following serum adaptation and the corresponding muropeptide profiles were analysed by rp-HPLC (Fig. 5c). When grown in TSB, no differences were detected between the WT and the *graS*::Tn mutant (Fig. 5c). As expected, incubation in serum resulted in additional peaks, likely resulting from the activity of human-derived peptidoglycan hydrolases present in serum[58]. However, these additional peaks were present in both the WT and

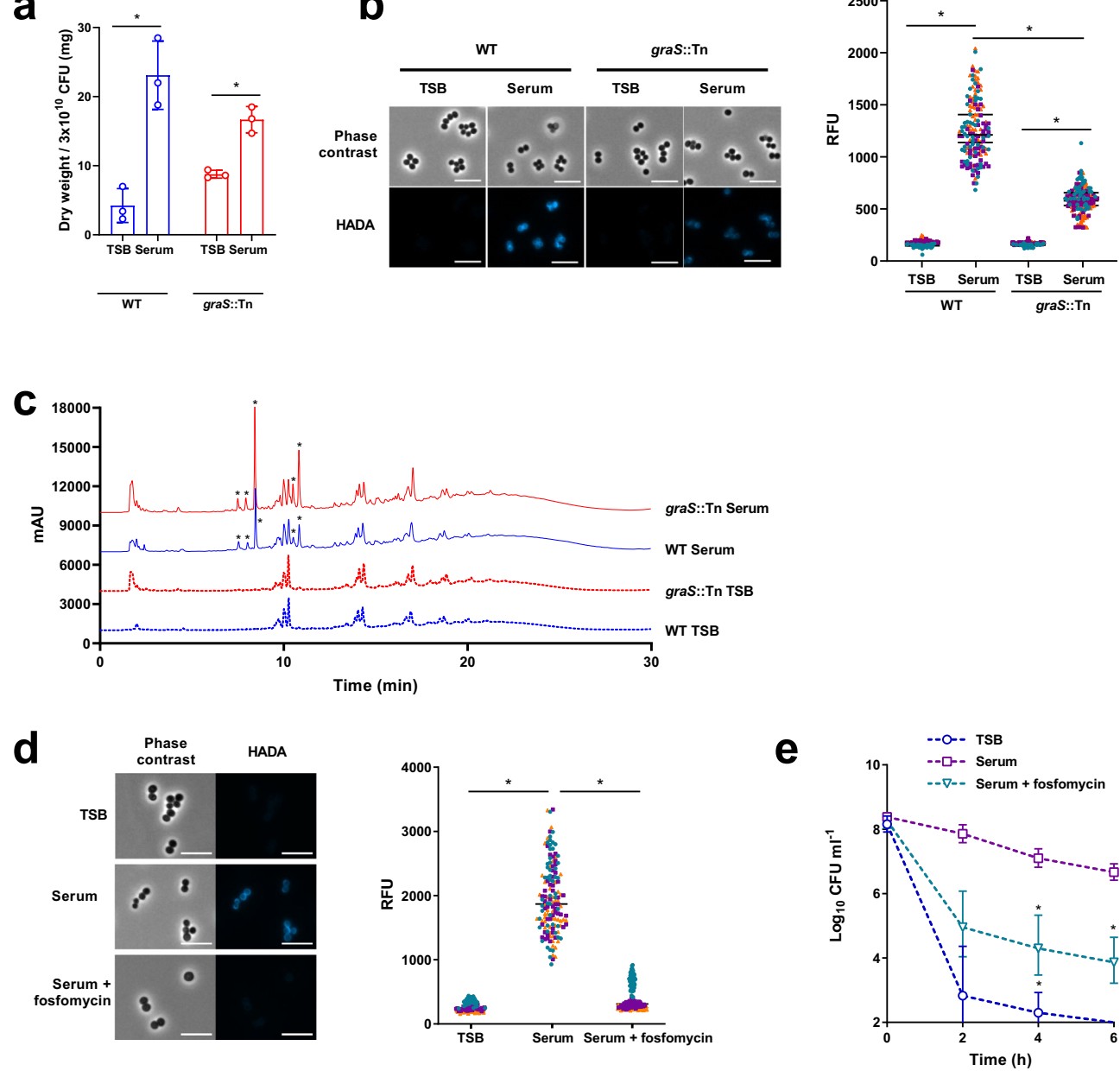

**Fig. 5 GraRS-mediated increase in peptidoglycan partially explains serum-induced tolerance.** Dry weight of peptidoglycan extracted from 300 ml cultures of TSB-grown or serum-adapted *S. aureus* JE2 WT or the *graS*::Tn mutant strain (**a**). Phase contrast and fluorescence microscopy of TSB-grown and serum-adapted cultures of *S. aureus* WT and the *graS*::Tn mutant strain (**b**). Scale bars, 5 µm. The fluorescence of individual cells was quantified. Graph represents the fluorescence of 50 cells per biological replicate (150 cells in total) with the mean of each replicate indicated. Each biological replicate is depicted in a different colour. Peptidoglycan from panel (**a**) was analysed by rp-HPLC (**c**). Asterisks denote peaks present in serum-adapted but not TSB-grown samples. Phase contrast and fluorescence microscopy of TSB-grown, serum-adapted or serum + 64 µg ml$^{-1}$ fosfomycin-adapted cultures of *S. aureus* JE2 WT (**d**). Scale bars, 5 µm. The fluorescence of individual cells was quantified and different colours represent different biological replicates. Log$_{10}$ CFU ml$^{-1}$ of bacteria depicted in panel (**d**) during a 6 h exposure to 80 µg ml$^{-1}$ daptomycin in serum (**e**). Graph in (**a**) represents the mean ± standard deviation of three independent experiments and data were analysed by two-way ANOVA with Sidak's *post-hoc* test (*$P < 0.0001$ (WT), =0.0217 (*graS*::Tn)). Data in (**b**) and (**d**) were analysed by Kruskal–Wallis with Dunn's *post hoc* test (* for (**b**), $P < 0.0001$ (WT), <0.0001 (*graS*::Tn). For (**d**), $P < 0.0001$ (TSB vs Serum), $P < 0.0001$ (Serum vs Serum + fosfomycin)). Data in (**e**) represent the geometric mean ± geometric standard deviation of three independent experiments and were analysed by two-way ANOVA with Dunnett's *post-hoc* test (*, $P = 0.0062$ (4 h serum), 0.0379 (4 h serum + Fosfomycin), 0.0358 (6 h serum + fosfomycin); serum-adapted vs serum/fosfomycin-adapted at each time-point). The final time point for TSB-grown cells fell below the limit of detection of 100 CFU ml$^{-1}$. CFU, colony-forming units; TSB, tryptic soy broth. Source data are provided as a Source Data file.

*graS*::Tn mutant samples (Fig. 5c). Collectively, these data suggest that incubation in serum does not result in GraRS-mediated peptidoglycan modifications.

Finally, we investigated whether the increased peptidoglycan of serum-adapted cells contributed to daptomycin tolerance. To do this, *S. aureus* were incubated in serum supplemented with a sub-lethal concentration of fosfomycin, a peptidoglycan synthesis inhibitor (Supplementary Fig. 14). As expected, this prevented the increase in HADA fluorescence triggered by serum (Fig. 5d). In line with earlier results, serum adaptation resulted in daptomycin tolerance in untreated bacteria (Fig. 5e). By contrast, supplementation of serum with fosfomycin reduced tolerance, with ~200-fold fewer CFU surviving 6 h daptomycin exposure (Fig. 5e).

Taken together, these findings demonstrated that serum adaptation conferred tolerance via a GraRS-dependent increase in peptidoglycan content.

**Cell wall accumulation is associated with lower hydrolytic activity**. The total amount of peptidoglycan depends on the balance between cell wall synthesis and degradation. As GraRS regulates several peptidoglycan hydrolases[53,56], we hypothesised that incubation in serum led to cell wall accumulation in part via a reduction in the rate of cell wall degradation. To test this, we measured rates of Triton-X-mediated autolysis. This showed that TSB-grown *S. aureus* lysed rapidly on exposure to Triton X-100, resulting in 37% of the initial optical density remaining after 6 h (Fig. 6a). By contrast, serum-adapted bacteria underwent significantly less lysis, with 93% of the initial optical density remaining after 6 h (Fig. 6a). Next, we carried out zymography to determine the abundance and activity of the hydrolytic enzymes present in the cell wall. Several bands corresponding to enzymes with hydrolytic activity were visible in the TSB-grown sample, whereas there were fewer and weaker bands in cell walls isolated from serum-adapted samples (Fig. 6b). This reduced hydrolytic activity was dependent on GraRS as serum-adapted cultures of the *graS*::Tn mutant underwent significantly more Triton-X-triggered lysis than the WT strain (Fig. 6c). This difference in lysis was complemented by the presence of the P*graXRS* plasmid but not by empty pCN34 (Fig. 6c). Additionally, zymography demonstrated that there were more hydrolases present in the sample from serum-adapted cultures of the *graS*::Tn mutant than the WT (Fig. 6d). These bands were also present in the wall extracts of the mutant strain complemented with empty pCN34 but were not observed when the mutant was complemented with P*graXRS* (Fig. 6d).

Taken together, serum-adapted *S. aureus* showed a GraRS-dependent decrease in hydrolytic activity compared to TSB-grown cultures, suggesting that the serum-induced accumulation of peptidoglycan was partly due to a reduction in the rate at which the cell wall was being degraded.

**Full daptomycin tolerance requires changes to the membrane**. While GraRS-dependent changes to peptidoglycan contributed to serum-induced daptomycin tolerance, they did not fully explain it. As the cell membrane is an important determinant of daptomycin susceptibility, we examined whether tolerance was also due to changes in membrane phospholipid composition. Phospholipids were extracted from TSB-grown and serum-adapted cultures using a method based on that of Bligh and Dyer[59]. However, as it has been shown that incubation in serum leads to host phospholipids associating with the staphylococcal membrane[35], cells were washed with Triton X-100 before extraction to remove these non-covalently bound lipids. After phospholipid extraction, thin layer chromatography (TLC) was used to separate phospholipids, which were then visualised with phosphoric acid and copper

sulphate[60]. As expected, this analysis showed the three main phospholipid species present in the staphylococcal membrane, cardiolipin (CL), PG and LPG[61]. Quantification showed that serum-adapted *S. aureus* contained significantly more CL (10.3% total phospholipids vs 7.8% in TSB-grown cells) and less PG (80% in TSB vs 75% in serum) than TSB-grown bacteria (Fig. 7a). Similar changes in phospholipid composition were observed on incubation of the *graS*::Tn mutant in serum, indicating that these changes were not GraRS-dependent (Fig. 7b).

To investigate whether this increase in CL was required for daptomycin tolerance, TSB-grown and serum-adapted cultures of JE2 WT and mutants in the two CL synthases present in *S. aureus* (*cls1*::Tn and *cls2*::Tn) were exposed to daptomycin and survival measured. There were no significant differences between the killing kinetics of the strains when TSB-grown, demonstrating that the susceptibilities of the strains to daptomycin were similar (Fig. 7c). As expected, serum adaptation conferred a high level of tolerance on the WT strain, with 58% of the population surviving 6 h daptomycin exposure (Fig. 7c). Similarly, the *cls1*::Tn mutant also showed high levels of tolerance (Fig. 7c). By contrast, serum adaptation did not protect the *cls2*::Tn mutant from daptomycin, with only 0.4% of the inoculum surviving after 6 h (Fig. 7c). This defect in daptomycin tolerance was fully complemented by the expression of *cls2* on a plasmid from its native promoter (Supplementary Fig. 15). As with the WT strain, incubation in serum led to a significant increase in the CL content of the *cls1*::Tn strain and a decrease in PG (Fig. 7d). By contrast, incubation in serum did not significantly affect the membrane composition of the *cls2*::Tn mutant strain, with no differences observed in the content of CL, PG or LPG between TSB-grown and serum-adapted cultures (Fig. 7e). Therefore, in agreement with its requirement for tolerance, Cls2, but not Cls1, was required for the accumulation of CL that occurred during serum adaptation.

While the *cls2*::Tn mutant strain showed significantly reduced tolerance compared to WT, tolerance was not completely abolished in this strain (Fig. 7c). Therefore, the final objective was to determine whether the increase in peptidoglycan and CL together fully explained daptomycin tolerance. To do this, peptidoglycan synthesis was inhibited in the WT strain with fosfomycin, CL accumulation was prevented by using the *cls2*::Tn mutant, or synthesis of both molecules was inhibited together by incubating the *cls2*::Tn mutant in serum supplemented with a sub-lethal concentration of fosfomycin. Cultures were then exposed to daptomycin for 6 h and survival measured. Inhibition of either cell wall synthesis or CL accumulation led to small reductions in tolerance (Fig. 7f). Inhibition of both peptidoglycan and CL accumulation together completely abolished daptomycin tolerance, with daptomycin killing these bacteria as efficiently as TSB-grown cultures (Fig. 7f).

Taken together, incubation in serum led to an increase in both peptidoglycan and CL, and inhibition of the synthesis of both these molecules completely prevented daptomycin tolerance.

## Discussion

Treatment of invasive MRSA infections is challenging due to limited effective treatment options. This means that infections are associated with high rates of treatment failure even when they are caused by drug-susceptible strains[14]. Understanding the reasons behind this failure is crucial to improving the efficacy of antibiotic therapy. There is evidence that, in some cases, treatment failure may be due to antibiotic tolerance being induced by the host environment, but unfortunately, this tolerance is not observed in standard laboratory media[14]. Using human serum to model bloodstream infections, we demonstrated that adaptation to the

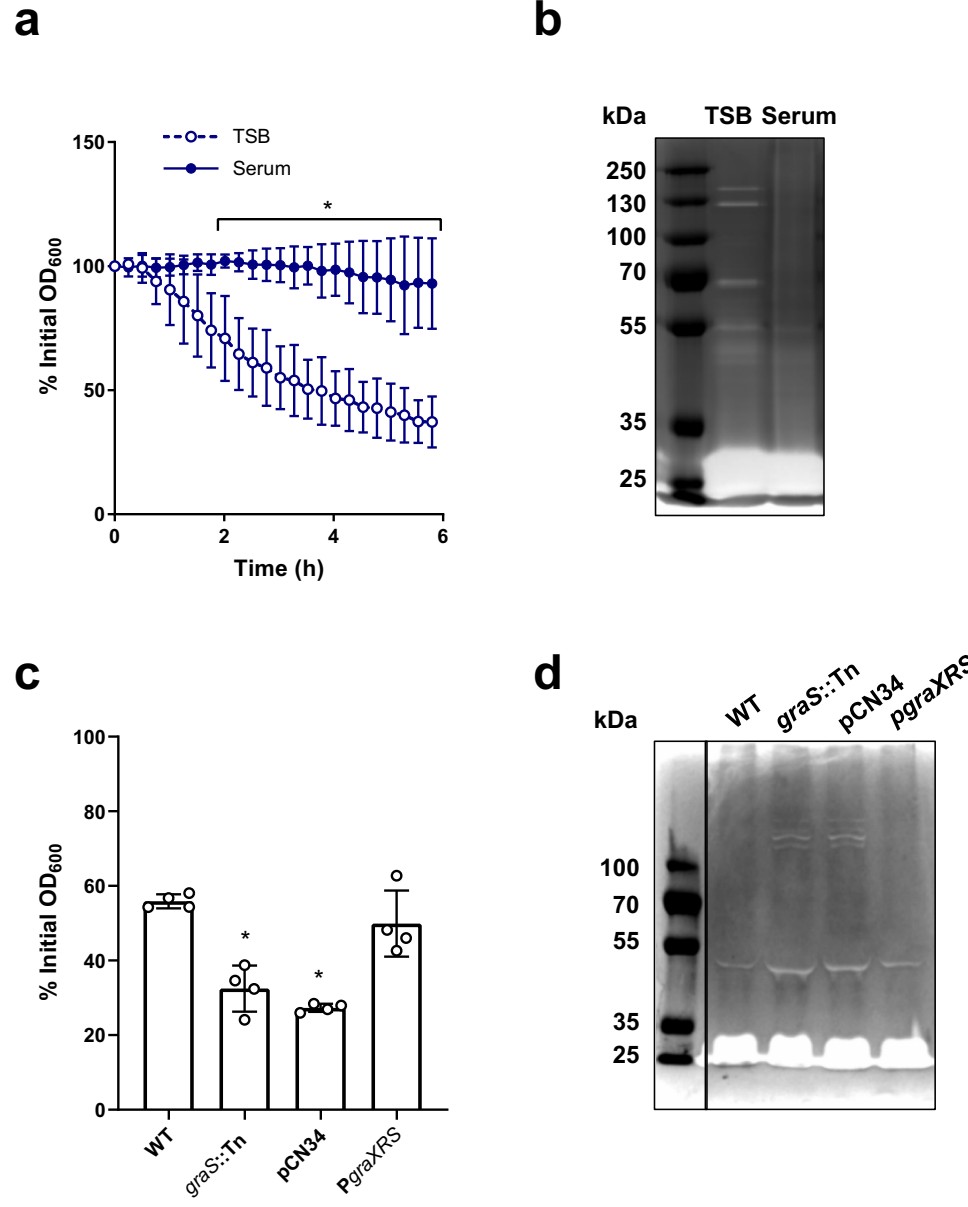

**Fig. 6 Accumulation of peptidoglycan is associated with a GraRS-dependent reduction in autolysis.** Triton X-100-triggered lysis of TSB-grown and serum-adapted cultures of *S. aureus* as measured by following $OD_{600}$ values during a 6 h exposure to 0.05% Triton X-100 (**a**). Cell wall extracts of TSB-grown and serum-adapted *S. aureus* were separated by SDS-PAGE using gels containing heat-killed *S. aureus* cells. Hydrolases were allowed to refold and zones of peptidoglycan degradation were visualised using methylene blue (**b**). Triton X-100-triggered lysis of serum-adapted cultures of *S. aureus* WT, the *graS*::Tn mutant, and the *graS*::Tn mutant complemented with empty pCN34 or P*graXRS* as measured by $OD_{600}$ values after a 6 h exposure to 0.05% Triton X-100 (**c**). Cell wall extracts of serum-adapted cultures of *S. aureus* WT, the *graS*::Tn mutant, and the *graS*::Tn mutant complemented with empty pCN34 or P*graXRS* were analysed by zymography (**d**). Graphs in (**a**) and (**c**) represent the mean ± standard deviation of three or four independent experiments, respectively. Three independent replicates of zymography were carried out and one representative image is shown. Data in (**a**) were analysed by two-way ANOVA with Sidak's *post-hoc* test; *, $P = 0.0267$ (120 min), 0.0031 (135 min), 0.0011 (150 min), 0.0005 (165 min) and <0.0001 at all time-points from 180 min onwards. Data in (**c**) were analysed by one-way ANOVA with Dunnett's *post-hoc* test; WT vs mutants (* $P = 0.0002$ (*graS*::Tn), <0.0001 (pCN34)). TSB, tryptic soy broth; $OD_{600}$, optical density at 600 nm. Source data are provided as a Source Data file.

host environment induced high levels of tolerance to several bactericidal antibiotics with different mechanisms of action, including vancomycin, gentamicin, nitrofurantoin and the last-resort antibiotic daptomycin.

Serum-triggered daptomycin tolerance was due in part to activation of GraRS by host LL-37, a finding that adds to our growing understanding that triggering of stress responses by the host can enable bacteria to survive antibiotic exposure[14]. LL-37 has previously been identified to promote transient tolerance of *S.*

*aureus* towards vancomycin, contributing to treatment failure in a *Galleria* infection model[31]. Furthermore, colistin exposure has been shown to enable *S. aureus* to grow in the presence of otherwise inhibitory concentrations of vancomycin[25]. Whilst it was not established if vancomycin tolerance was due to activation of GraRS by LL-37 or colistin, there were phenotypic similarities to what we observed, including increased cell wall thickness[25,31].

Serum-induced daptomycin tolerance was due to changes in both the cell wall and the cell membrane, which together

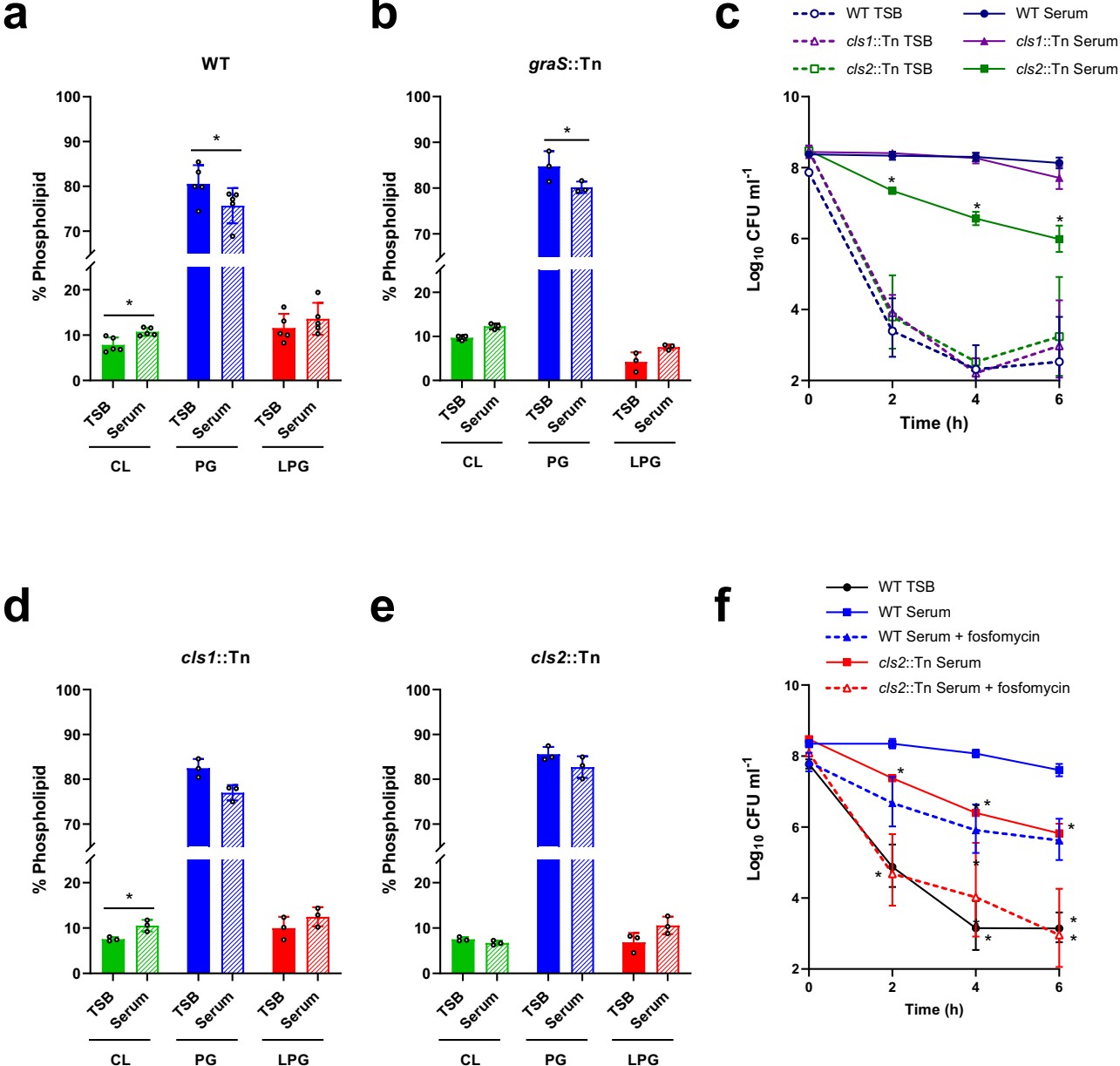

**Fig. 7 GraRS-independent changes to the staphylococcal membrane also contribute to tolerance.** Phospholipids were extracted from TSB-grown and serum-adapted cultures of *S. aureus* JE2 WT (**a**), the *graS*::Tn mutant (**b**), the *cls1*::Tn mutant (**d**) and the *cls2*::Tn mutant (**e**) before being analysed by thin layer chromatography and visualised with copper sulphate and phosphoric acid. The relative phospholipid compositions were determined by quantifying spot intensities using ImageJ. $Log_{10}$ CFU $ml^{-1}$ of TSB-grown and serum-adapted cultures of *S. aureus* JE2 WT and the *cls1*::Tn and *cls2*::Tn mutant strains during a 6 h exposure to 80 µg $ml^{-1}$ daptomycin in serum (**c**). $Log_{10}$ CFU $ml^{-1}$ of TSB-grown and serum-adapted cultures of *S. aureus* JE2 WT and the *cls2*::Tn mutant strains during a 6 h exposure to 80 µg $ml^{-1}$ daptomycin in serum (**f**). Where appropriate, adaptation was carried out with sub-lethal concentrations of fosfomycin (dashed lines). Data in (**a**), (**b**), (**d**) and (**e**) represent the mean ± standard deviation of at least three independent experiments and were analysed by paired *t*-tests (* for (**a**), $P = 0.0015$ (CL), 0.0116 (PG), 0.1002 (LPG). For (**b**) P = 0.2642 (CL), 0.0222 (PG), 0.1069 (LPG). For (**d**), $P = 0.0491$ (CL); TSB-grown vs serum-adapted for each phospholipid. Data in (**c**) and (**f**) represent the geometric mean ± geometric standard deviation of three independent replicates and were analysed by two-way ANOVA with Dunnetts's *post-hoc* test * for (**c**), $P = 0.0008$ (2 h), 0.001 (4 h) and 0.0066 (6 h) serum-adapted WT vs serum-adapted mutant. For (**f**), $P = 0.0163$ (2 h WT TSB), 0.0115 (4 h WT TSB), 0.0018 (6 h WT TSB), 0.0022 (*cls2*::Tn Serum 2 h), 0.0002 (*cls2*::Tn Serum 4 h), 0.0034 (*cls2*::Tn Serum 6 h), 0.0349 (*cls2*::Tn Serum + fosfomycin 6 h). CL, cardiolipin; PG, phosphatidylglycerol; LPG, lysyl-phosphatidylglycerol; TSB, tryptic soy broth; CFU, colony-forming units. Source data are provided as a Source Data file.

appeared to significantly reduce daptomycin binding to its membrane targets of PG and cell wall precursors such as lipid II[3–5]. Furthermore, the increased abundance of WTA and peptidoglycan may have depleted lipid II, reducing daptomycin membrane targets, whilst accumulation of CL would be expected to impair the antibacterial effect of the small amount of

daptomycin that did bind cells by preventing the antibiotic from translocating the membrane[62].

The serum-induced increase in peptidoglycan was dependent on GraRS, partly due to a decrease in cell wall degradation as incubation in serum led to a GraRS-dependent decrease in the abundance and activity of peptidoglycan hydrolases. Importantly,

this increase in peptidoglycan is in line with the results of other studies that have investigated the differences between *S. aureus* grown in culture medium and in model host environments. For example, Hines et al[35] demonstrated that the cell walls of *S. aureus* grown in 20% human serum were twice as thick as those grown in TSB, while Sutton et al[37] observed that the cell walls of *S. aureus* isolated from an experimental murine kidney abscess model were significantly thicker than those of cells grown in vitro. Therefore, we are confident that the results from our ex vivo model system are applicable to the host environment during infection. Furthermore, GraRS was found to contribute to *S. aureus* virulence in both a murine and a *Galleria mellonella* model system, providing evidence that this system is activated during infection[63,64].

As LL-37 is produced by both macrophages and neutrophils it is likely to be present at many sites of bacterial infection and may therefore contribute to antibiotic tolerance at diverse anatomical sites as well as the bloodstream[65]. The concentration of synthetic LL-37 required here to induce daptomycin tolerance is greater than concentrations reported in healthy human serum[66,67]. A super-physiological concentration may be required because synthetic LL-37 may be less active than the native peptide or because the peptide may be degraded by *S. aureus*. Several proteases produced by *S. aureus*, including aureolysin and V8 protease, can degrade LL-37[68]. It may also be the case that LL-37 is a more potent trigger of GraRS in serum than RPMI 1640.

An additional consideration is that whilst low concentrations of LL-37 are present in the serum of healthy individuals, this concentration is much higher during infection and can be extremely high locally at infection sites[66,69,70]. Furthermore, neutrophil granules have been proposed to contain as much as 180 μg ml$^{-1}$ LL-37[67]. Therefore, this high local LL-37 concentration, combined with GraRS activation due to the acidic pH inside the phagosome[71,72] may promote daptomycin tolerance inside this intracellular compartment. These surviving bacteria may then be spread around the body by circulating neutrophils[73], seeding secondary infections and worsening patient prognoses.

Sensing of LL-37 by *S. aureus* is important because this AMP has anti-staphylococcal activity and mutants lacking GraRS are more susceptible to LL-37 and have decreased survival in a murine infection model[63]. Therefore, we hypothesise that GraRS mediated sensing of LL-37 and the resulting cell wall thickening constitutes a defence against LL-37 and other AMPs. In support of this, we found that serum triggered tolerance of the AMPs nisin and gramicidin, as well as daptomycin.

The importance of the peptidoglycan layer in influencing daptomycin susceptibility fits with observations that the thickened cell walls of vancomycin-intermediate *S. aureus* (VISA) strains frequently confer reduced susceptibility to daptomycin[74,75]. Additionally, studies of daptomycin non-susceptible (DNS) isolates revealed thicker cell walls in many cases and metabolic studies of DNS isolates demonstrated a shift from glycolysis towards the pentose phosphate pathway which is used to generate precursors for peptidoglycan synthesis[76]. Furthermore, Bertsche et al observed[77,78] that DNS strains contained higher levels of both WTA and peptidoglycan than susceptible isolates, however, they did not determine which polymer was responsible for the resistance. We found that D-alanylated WTA was necessary for peptidoglycan accumulation and thus tolerance, possibly because WTA has been shown to protect peptidoglycan from degradation by the autolysin Atl[79].

Although serum adaptation resulted in increased WTA and D-alanine content in the cell wall relative to TSB-grown cells, this accumulation did not contribute to tolerance directly, rather it was the WTA-enabled accumulation of peptidoglycan that was responsible. This discovery that it is the peptidoglycan rather than

the WTA component of the wall which mediates tolerance may inform about the mechanisms used by *S. aureus* to survive daptomycin exposure more broadly.

In addition to changes to the cell wall, serum also induced daptomycin tolerance through alterations in phospholipid composition, specifically, a Cls2-dependent increase in CL and decrease in PG content. Both decreased PG and increased CL have previously been associated with the DNS phenotype[80]. As PG is a target of daptomycin, it is clear why a reduction in this phospholipid may mediate tolerance. Furthermore, an increased CL content in some DNS isolates has been observed to increase membrane rigidity and thickness, impairing daptomycin penetration[80] and thus access to lipid-coupled cell wall precursors.

Accumulation of CL by *S. aureus* in serum was due to Cls2 rather than Cls1. Previous work has shown that Cls2 is the 'housekeeping' CL synthase and responsible for CL accumulation in stationary phase[81]. As such, it is possible that the bacteriostasis conferred upon *S. aureus* by serum mimics stationary phase, leading to Cls2-mediated CL accumulation. Cls2, together with Cls1, is also required for CL accumulation in response to phagocytosis by neutrophils, although it is not known whether LL-37 produced by phagocytic cells triggers this phenomenon[81].

The large differences in TSB-grown and serum-adapted cell envelopes indicate that tolerant cells are in a very different state from drug-susceptible bacteria, suggesting that they have undergone significant changes in transcription and metabolism. These findings are in line with those of Peyrusson et al[27], who studied the transcriptional responses of *S. aureus* persisters inside infected host cells. They found that persisters were metabolically active and showed altered transcription characteristic of the activation of several stress responses, including the stringent response, cell wall stress response and SOS response[27]. Therefore, as is the case when exposed to serum, *S. aureus* can be in a non-replicative state but maintain metabolic and transcriptional activity, highlighting the difference between a lack of growth and a lack of metabolism and demonstrating that metabolic inactivity is not required for daptomycin tolerance.

Whilst we identified two independent mechanisms by which serum triggered daptomycin tolerance, we do not yet know how serum adaptation confers tolerance to the other antimicrobials tested. The similarities of serum-adapted *S. aureus* to VISA isolates may explain vancomycin tolerance and we hypothesize that the thickened cell wall may exclude the AMPs nisin and gramicidin from reaching their membrane targets[82,83]. Furthermore, the increased abundance of D-alanylated WTA in the walls of serum-adapted cells may contribute to gentamicin tolerance, since *dltA* has been shown to be required for survival of *Streptococcus mutans* biofilm exposed to this antibiotic[84]. However, we do not currently have a hypothesis for the mechanism responsible for nitrofurantion tolerance in serum-adapted cells.

Understanding the mechanisms behind daptomycin tolerance is important for developing a combination therapeutic approach using readily available drugs as a simple and effective way to improve treatment outcomes. Serum-induced tolerance occurred via two mechanisms: a GraRS-dependent accumulation of peptidoglycan and a GraRS-independent increase in CL. Inhibition of either of these processes significantly reduced the extent of daptomycin tolerance in vitro, indicating that targeting these processes may be a viable strategy to reduce tolerance. Unfortunately, although we demonstrated that inhibition of GraRS using verteporfin reduced daptomycin tolerance in serum, this drug is unsuitable for use systemically due to toxicity, and a second GraRS inhibitor, MAC-545496 cannot be used due to poor aqueous solubility[51,85]. Therefore, although inhibition of GraRS may reduce daptomycin tolerance, a lack of suitable inhibitors

currently precludes their testing in in vivo invasive disease models. Inhibition of CL accumulation is also not currently a viable approach as, to the best of our knowledge, no inhibitors of *S. aureus* Cls2 have been described. By contrast, the finding that peptidoglycan synthesis was essential for host-mediated daptomycin tolerance raises the possibility that combined use of daptomycin with an additional cell wall synthesis inhibitor may prevent tolerance. In support of this, supplementation of serum with fosfomycin prevented cell wall accumulation and significantly reduced the number of bacteria which survived subsequent daptomycin exposure. As well as preventing tolerance, a combination approach may also have other benefits as fosfomycin has been shown to synergise with daptomycin in in vitro and in vivo studies and to slow the development of daptomycin resistance[86–90]. The basis of this synergy is thought to be an increase in daptomycin binding, providing additional evidence that the wall acts as a significant barrier for daptomycin and that by reducing cell wall thickness daptomycin can reach its membrane target more easily[91].

Further evidence to support the clinical use of daptomycin and fosfomycin in combination comes from a recently-reported randomised trial of patients with MRSA bacteraemia[6]. This found that the combination of daptomycin and fosfomycin was associated with 12% higher rate of treatment success than daptomycin monotherapy[6].

In summary, we have we have identified a mechanism by which human serum triggers antibiotic tolerance in *S. aureus*, including towards the last-resort antibiotic daptomycin. This provides a potential explanation for the relatively low efficacy of daptomycin when used in patients, which contrasts with its potent bactericidal activity in vitro. Daptomycin tolerance occurred via two distinct mechanisms: a GraRS-dependent increase in peptidoglycan and an increase in the CL content of the membrane. Inhibition of either of these processes reduced the development of tolerance, providing a rationale for a combination therapeutic approach to prevent daptomycin tolerance and enable the lipopeptide antibiotic to kill *S. aureus* effectively.

## Methods

**Bacterial strains and growth conditions**. The *S. aureus* strains used in this study are listed in Table 1. *S. aureus* was cultured in 3 ml tryptic soy broth (TSB) in 30 ml universal tubes and grown for 15 to 17 h at 37 °C with shaking (180 rpm) to reach stationary phase. When necessary, TSB was supplemented with 10 μg ml$^{-1}$ erythromycin or 90 μg ml$^{-1}$ kanamycin. Since daptomycin requires calcium for activity, laboratory media were supplemented with a final concentration of 50 μg ml$^{-1}$ Ca$^{2+}$. Since the serum used has physiological concentrations of calcium (44–56 μg ml$^{-1}$)[92] this was not supplemented with calcium.

To generate TSB-grown bacteria, TSB was inoculated with 10$^7$ CFU ml$^{-1}$ from overnight cultures and incubated at 37 °C with shaking (180 rpm) until 10$^8$ CFU ml$^{-1}$ was reached (2–2.5 h depending on strain). To generate serum-adapted bacteria, these TSB-grown bacteria were centrifuged, resuspended in human serum from male AB plasma (Sigma) and incubated for 16 h at 37 °C with shaking (180 rpm). 16 h was chosen as a reasonable estimate for the time from infection onset to treatment. We allowed 4 h for symptom onset and 12 h for diagnosis by blood culture[93]. To mitigate differences between batches of serum, each experiment was performed with a single batch of human serum.

As *S. aureus* is unable to replicate in human serum, bacterial CFU counts of serum-adapted cultures were equal to those of TSB-grown cultures. Where necessary, serum was supplemented with sub-lethal concentrations of additional antibiotics: tunicamycin (128 μg ml$^{-1}$, Abcam), targocil (128 μg ml$^{-1}$, Cambridge Bioscience), tarocin A1 (64 μg ml$^{-1}$, Sigma), fosfomycin (64 μg ml$^{-1}$, Tokyo Chemical Industry) or verteporfin (10–80 μg ml$^{-1}$; Sigma). Where appropriate, serum was incubated for 2 h with polyclonal sheep IgG anti-LL-37 (2.5–20 μg ml$^{-1}$; R&D Systems) or anti-hNP-1 (2.5–20 μg ml$^{-1}$; R&D Systems) antibodies before serum-adaptation was carried out.

Where appropriate, adaptation was carried out in serum fractions. These were generated by dialysis using membranes with various molecular weight cut-offs (0.5 kDa (Float-a-lyser G2; Spectrum Labs); 5 kDa (SnakeSkin dialysis tubing; Thermo Fisher) or 100 kDa (Float-a-lyser G2; Spectrum Labs)). Whole serum (2 ml) was dialysed at 4 °C for 24 h against 3 changes of 400 ml distilled water. To generate the <5 kDa fraction, 1 ml serum was dialysed against 3 rounds of 3 h in

20 ml changes of distilled H$_2$O at 4 °C. The dialysate was collected, frozen with liquid nitrogen and lyophilised. The resulting powder was resuspended in 1 ml distilled H$_2$O and used for adaptation.

RPMI-adapted bacteria were generated by resuspending TSB-grown cultures in an equal volume of RPMI 1640 and incubating for 16 h at 37 °C with shaking (180 rpm). Where necessary, RPMI 1640 was supplemented with colistin (2.5–20 μg ml$^{-1}$, Sigma), LL-37 (5–40 μg ml$^{-1}$, R&D Systems), hNP-1 (0.625–10 μg ml$^{-1}$, Cambridge Bioscience) or dermcidin (12.5–100 μg ml$^{-1}$, Cambridge Bioscience).

**Determination of antibiotic MICs**. MICs were determined by broth microdilution as described previously[94]. Two-fold serial dilutions of antibiotics in TSB were generated in flat-bottomed 96-well plates. Wells were inoculated to a final concentration of 5 × 10$^5$ CFU ml$^{-1}$ stationary phase *S. aureus* and incubated for 16 h at 37 °C under static conditions. The MIC was defined as the lowest concentration at which no visible growth was observed.

**Antibiotic killing assays**. 3 ml cultures of 10$^8$ CFU ml$^{-1}$ TSB-grown and serum/RMPI-adapted *S. aureus* were generated as described above. TSB-grown cultures were centrifuged and resuspended in human serum or RPMI 1640 immediately before the assay. When antibiotic killing assays were performed in RPMI 1640, CaCl$_2$ was added to a final concentration of 1.25 mM. The antibiotic or antimicrobial peptide of interest was added to these TSB-grown bacteria or to serum-adapted *S. aureus*. Cultures were incubated at 37 °C with shaking (180 rpm). At each time-point, aliquots were taken, serially diluted 10-fold in PBS in 96-well plates and plated to determine bacterial viability by CFU counts. Survival was calculated as a percentage of the starting inoculum.

**Labelling of daptomycin with BoDipy fluorophore**. Labelling of daptomycin with the BoDipy fluorophore was performed as described previously[95,96]. The primary amine group of daptomycin was used conjugate the antibiotic to the BoDipy fluorophore via its NHS ester. 100 μl 50 mg ml$^{-1}$ daptomycin was incubated with 50 μl 10 mg ml$^{-1}$ BoDipy FL SE (Thermo Fisher Scientific) and 850 μl 0.2 M sodium bicarbonate (pH 8.5) for 4 h at 37 °C. Unconjugated BoDipy was removed by dialysis against water at 4 °C using a Float-A-Lyzer G2 device with a 0.5 kDa molecular weight cut-off.

**Measurements of daptomycin binding to *S. aureus***. To measure daptomycin binding to *S. aureus*, 3 ml TSB-grown or serum-adapted bacteria were generated as described above and incubated at 37 °C (shaking at 180 rpm) with 320 μg ml$^{-1}$ BoDipy-labelled daptomycin in serum. Every 2 h, aliquots were taken, washed three times in PBS, and then fluorescence was measured using a TECAN Infinite 200 PRO microplate reader (excitation 490 nm; emission 525 nm). Fluorescence values were blanked against uninoculated wells and divided by OD$_{600}$ to normalise for cell density.

**Fluorescence and phase-contrast microscopy**. For measurements of daptomycin binding, TSB-grown or serum-adapted *S. aureus* were generated and incubated with BoDipy-daptomycin as described above. Samples were incubated with 10 μg ml$^{-1}$ nile red for 10 mins at 37 °C to visualise cell membranes. Samples were washed 3 times in PBS and fixed in 4% paraformaldehyde. For measurements of peptidoglycan synthesis, TSB-grown and serum-adapted *S. aureus* were generated as described above, but with the addition of 25 μM HADA at each step. Samples were washed 3 times in PBS and fixed in 4% paraformaldehyde. For measurements of bacterial surface charge, 1 ml aliquots of TSB-grown and serum-adapted cultures of *S. aureus* were incubated with 80 μg ml$^{-1}$ FITC-PLL for 10 min at room temperature in the dark. Cells were then washed three times in PBS and fixed in 4% paraformaldehyde in the dark.

Aliquots (2 μl) of fixed bacteria were spotted onto microscope slides covered with a thin agarose layer (1.2% agarose in water) and covered with a cover slip. Phase-contrast and fluorescence images were taken using a Zeiss Axio Imager.A1 microscope coupled to an AxioCam MRm and a 100× objective and processed using the Zen 2012 software (blue edition). BoDipy-daptomycin and FITC-PLL were detected using a green fluorescent protein filter set, Nile red fluorescence signals were detected using a Texas Red filter set, and HADA using a DAPI filter set. Microscopy of all samples within an experiment were performed at the same time using identical settings to allow comparisons to be made between samples. The fluorescence intensities of individual cells were quantified using Zen 2012 software (blue edition).

**Determination of membrane polarity**. Membrane polarity was measured using 3,3′-dipropylthiadicarbocyanine iodide (DiSC$_3$(5), Thermofisher Scientific) as described previously[46] with some modifications. 3 ml cultures of TSB-grown or serum-adapted *S. aureus* were generated as described above and incubated with 80 μg ml$^{-1}$ daptomycin in human serum at 37 °C with shaking (180 rpm). After 6 h daptomycin exposure, 200 μl aliquots of cultures were added to black flat-bottomed 96-well plates. DiSC$_3$(5) was added to a final concentration of 1 μM, mixed and the plate incubated statically at 37 °C for 5 min. Fluorescence was then measured using

**Table 1 Bacterial strains.**

| Strain | Description | Reference or source |
|---|---|---|
| *Staphylococcus aureus* | | |
| USA300 LAC* | LAC strain of the USA300 CA-MRSA lineage, cured of LAC-p03 plasmid | 104 |
| USA300 LAC* Δ*dltD* | USA300 LAC* with the *dltD* gene deleted, Ery$^r$ | This study |
| USA300 LAC* Δ*dltD* pCN34 | USA300 LAC* Δ*dltD* carrying the empty pCN34 vector, Ery$^r$ Kan$^r$ | This study |
| USA300 LAC* Δ*dltD* P*dltD* | USA300 LAC* Δ*dltD* complemented with P*dltD*, Ery$^r$ Kan$^r$ | This study |
| USA300 LAC JE2 | LAC strain of the USA300 CA-MRSA lineage cured of plasmids | 47 |
| USA300 LAC JE2 *agrC*::Tn | USA300 LAC JE2 with a *bursa aurealis* transposon insertion in *agrC*, Ery$^r$ | 47 |
| USA300 LAC JE2 *saeS*::Tn | USA300 LAC JE2 with a *bursa aurealis* transposon insertion in *saeS*, Ery$^r$ | 47 |
| USA300 LAC JE2 *arlS*::Tn | USA300 LAC JE2 with a *bursa aurealis* transposon insertion in *arlS*, Ery$^r$ | 47 |
| USA300 LAC JE2 *lytS*::Tn | USA300 LAC JE2 with a *bursa aurealis* transposon insertion in *lytS*, Ery$^r$ | 47 |
| USA300 LAC JE2 *srrB*::Tn | USA300 LAC JE2 with a *bursa aurealis* transposon insertion in *srrB*, Ery$^r$ | 47 |
| USA300 LAC JE2 *nreB*::Tn | USA300 LAC JE2 with a *bursa aurealis* transposon insertion in *nreB*, Ery$^r$ | 47 |
| USA300 LAC JE2 *airS*::Tn | USA300 LAC JE2 with a *bursa aurealis* transposon insertion in *airS*, Ery$^r$ | 47 |
| USA300 LAC JE2 *hptS*::Tn | USA300 LAC JE2 with a *bursa aurealis* transposon insertion in *hptS*, Ery$^r$ | 47 |
| USA300 LAC JE2 *hssS*::Tn | USA300 LAC JE2 with a *bursa aurealis* transposon insertion in *hssS*, Ery$^r$ | 47 |
| USA300 LAC JE2 *kdpD*::Tn | USA300 LAC JE2 with a *bursa aurealis* transposon insertion in *kdpD*, Ery$^r$ | 47 |
| USA300 LAC JE2 *phoP*::Tn | USA300 LAC JE2 with a *bursa aurealis* transposon insertion in *phoP*, Ery$^r$ | 47 |
| USA300 LAC JE2 *vraS*::Tn | USA300 LAC JE2 with a *bursa aurealis* transposon insertion in *vraS*, Ery$^r$ | 47 |
| USA300 LAC JE2 *graS*::Tn | USA300 LAC JE2 with a *bursa aurealis* transposon insertion in *graS*, Ery$^r$ | 47 |
| USA300 LAC JE2 *braS*::Tn | USA300 LAC JE2 with a *bursa aurealis* transposon insertion in *braS*, Ery$^r$ | 47 |
| USA300 LAC JE2 *desK*::Tn | USA300 LAC JE2 with a *bursa aurealis* transposon insertion in *desK*, Ery$^r$ | 47 |
| USA300 LAC JE2 *graS*::Tn pCN34 | USA300 LAC JE2 *graS*::Tn carrying the empty pCN34 plasmid, Ery$^r$ Kan$^r$ | This study |
| USA300 LAC JE2 *graS*::Tn P*graXRS* | USA300 LAC JE2 *graS*::Tn complemented with P*graXRS*, Ery$^r$ Kan$^r$ | This study |
| USA300 LAC JE2 P*dltA-gfp* | USA300 LAC JE2 carrying the P*dltA-gfp* reporter plasmid, Kan$^r$ | This study |
| USA300 LAC JE2 *graS*::Tn P*dltA-gfp* | USA300 LAC JE2 *graS*::Tn carrying the P*dltA-gfp* reporter plasmid, Ery$^r$ Kan$^r$ | This study |
| USA300 LAC JE2 *vraS*:Tn pCN34 | USA300 LAC JE2 *vraS*::Tn carrying the empty pCN34 plasmid, Ery$^r$ Kan$^r$ | This study |
| USA300 LAC JE2 *vraS*:Tn P*vraUTSR* | USA300 LAC JE2 *vraS*::Tn complemented with P*vraUTSR*, Ery$^r$ Kan$^r$ | This study |
| USA300 LAC JE2 P*vraX-gfp* | USA300 LAC JE2 carrying the P*vraX-gfp* reporter plasmid, Kan$^r$ | This study |
| USA300 LAC JE2 *vraS*::Tn P*vraX-gfp* | USA300 LAC JE2 *vraS*::Tn carrying the P*vraX-gfp* reporter plasmid, Ery$^r$ Kan$^r$ | This study |
| USA300 LAC JE2 *mprF*::Tn | USA300 LAC JE2 with a *bursa aurealis* transposon insertion in *mprF*, Ery$^r$ | 47 |
| USA300 LAC JE2 *cls1*::Tn | USA300 LAC JE2 with a *bursa aurealis* transposon insertion in *cls1*, Ery$^r$ | 47 |
| USA300 LAC JE2 *cls2*::Tn | USA300 LAC JE2 with a *bursa aurealis* transposon insertion in *cls2*, Ery$^r$ | 47 |
| USA300 LAC JE2 *cls2*::Tn pCN34 | USA300 LAC JE2 *cls2*::Tn carrying the empty pCN34 vector, Ery$^r$ Kan$^r$ | This study |
| USA300 LAC JE2 *cls2*::Tn P*cls2* | USA300 LAC JE2 *cls2*::Tn complemented with P*cls2*, Ery$^r$ Kan$^r$ | This study |

a TECAN Infinite 200 PRO microplate reader (excitation 622 nm; emission 670 nm). Fluorescence values were divided by OD$_{600}$ measurements to normalise for cell density.

**Determination of membrane permeability.** Daptomycin-induced membrane permeability was measured using propidium iodide, a membrane-impermeant dye that fluoresces when bound to DNA, as described previously[97]. 3 ml cultures of TSB-grown or serum-adapted *S. aureus* were generated as described above and incubated with 80 μg ml$^{-1}$ daptomycin in human serum at 37 °C with shaking (180 rpm). Aliquots were taken after 6 h daptomycin exposure and washed 3 times in PBS. 200 μl was transferred to a black-walled 96-well microtitre plate and PI was added to a final concentration of 2.5 μM. Fluorescence was measured using a TECAN Infinite 200 PRO microplate reader (excitation 535 nm; emission 617 nm). Fluorescence values were blanked against values from uninoculated wells and divided by OD$_{600}$ to normalise for cell density.

**Construction of strains.** P*vraX-gfp* and P*dltA-gfp* plasmids were constructed using Gibson assembly to measure induction of VraSR and GraRS signalling respectively. To construct P*vraX-gfp*, the *vraX* promoter was amplified from JE2 WT genomic DNA using primers *vraX*_Fw and *vraX*_Rev (Table 2). The pCN34 vector, together with *gfp*, was amplified from a previously constructed reporter plasmid[98] that contained a *recA* promoter-*gfp* construct using primers pCN34_*gfp*_Fw_*vraX* and pCN34_*gfp*_Rev_*vraX*. To construct P*dltA-gfp*, the *dltA* promoter was amplified from JE2 WT genomic DNA using primers *dltA*_Fw and *dltA*_Rev. The pCN34 vector, together with *gfp*, was amplified from a previously constructed reporter plasmid that contained a *recA* promoter-*gfp* construct[98] using primers pCN34_*gfp*_Fw_*dltA* and pCN34_*gfp*_Rev_*dltA*. Promoters were inserted into pCN34 upstream of *gfp* using Gibson assembly. Plasmids were transformed into *E. coli* DC10B, then electroporated into *S. aureus* RN4220 and finally transduced using φ11 into JE2 WT and the relevant mutant strain.

To construct the USA300 Δ*dltD* deletion mutant strain the *dltD* gene was replaced with an *erm* resistance marker in RN4220 Δ*spa*[99]. A phage 85 lysate was made of the RN4220 Δ*spa* Δ*dltD* strain and used to transduce USA300. Deletion of *dltD* was confirmed by PCR and sequencing.

**Fluorescent reporter assays.** Promoter-*gfp* constructs were used to measure gene expression in response to serum and LL-37 (5–80 μg ml$^{-1}$). To determine the response to serum, TSB-grown cultures were resuspended in serum and incubated at 37 °C. with shaking (180 rpm) in the dark for 6 h. At each time-point, 100 μl aliquots were removed, washed twice in PBS, resuspended in 100 μl PBS and transferred to a black 96-well plate. Fluorescence was measured at each time-point, and values were blanked against the fluorescence values of strains lacking the reporter construct.

To determine the response to LL-37, TSB-grown cultures (10 ml) were generated and resuspended in 100 μl PBS, resulting in a cell density of 10$^{10}$ CFU ml$^{-1}$. A black-walled 96-well plate was prepared, with each well containing 200 μl RPMI 1640 supplemented with a range of concentrations of LL-37. Bacteria (2 μl) were inoculated into each well, resulting in an inoculum of 10$^8$ CFU ml$^{-1}$.

The same approach was used to measure reporter activity in response to synthetic dermcidin (Cambridge Bioscience) or activated platelet supernatant from healthy donors. To generate this, whole human blood (30 ml) was collected in sodium citrate tubes (BD biosciences) from healthy male (2) and female (1) donors aged between 25 and 35 years. Platelet rich plasma (PRP) was separated from red blood cells by centrifugation at 200 × *g* for 10 min. A second round of centrifugation 600 × *g* for 10 min collected the platelets, which were resuspended in RPMI 1640 (Thermo Fisher). CaCl$_2$ was added to a final concentration of 1.25 mM and platelets were incubated at 37 °C, with shaking (180 rpm) for 1 h until aggregation occurred. Tubes were centrifuged at 1000 × *g* for 10 min to remove the activated platelets and the supernatant was centrifuged again at 1000 × *g* for 10 min. The resulting supernatant was stored at −80 °C. Ethical approval for drawing and processing human blood was obtained from the Regional Ethics Committee of Imperial College healthcare tissue bank (Imperial College London) and the Imperial NHS Trust Tissue Bank (REC Wales approval no. 12/WA/0196 and ICHTB HTA license no. 12275). Written informed consent was obtained from the donors prior to taking samples.

Plates containing peptide-exposed bacteria were inserted into a TECAN Infinite M200 PRO microplate reader and incubated at 37 °C for 16 h with orbital shaking (300 rpm). OD$_{600}$ and fluorescence intensity (excitation 485 nm; emission 525 nm) were measured every 15 min. Fluorescence values were blanked against the fluorescence values of strains lacking the reporter construct. Blanked fluorescence

**Table 2 Primers.**

| Oligonucleotide | Sequence (5′-3′) |
| --- | --- |
| For construction of PvraX-gfp fluorescent reporter plasmid: | |
| pCN34_gfp_Fw _vraX | AGCAAAGGAGGTAATATAGGAAAAAAAATGAGTAAAGGAGAAGAACTTTTCACTGG |
| pCN34_gfp_Rev_vraX | GTTGTATGCACCGTGATCCAGTGTCATCTGAACCATAGGATCCTCT |
| vraX_Fw | TCCTATGGTTCAGATGACACTGGATCACGGTGCATACAACCG |
| vraX_rev | CTCCTTTACTCATTTTTTTTCCTATATTACCTCCTTTGCTACTCTATGG |
| For construction of PdltA-gfp fluorescent reporter plasmid: | |
| pCN34_gfp_Fw_dltA | TCTAATGAGGGAGACTTAATAAAAAAAATGAGTAAAGGAGAAGAAC |
| pCN34_gfp_Rev_dltA | AATTATCATCAGCGCAAATAGTGTCATCTGAACCATAGGA |
| dltA_Fw | TCCTATGGTTCAGATGACACTATTTGCGCTGATGATAATTCA |
| dltA_Rev | TCTCCTTTACTCATTTTTTTTATTAAGTCTCCCTCATTAGAACT |
| For construction of PvraUTSR complementation plasmid: | |
| pCN34_Fw_vra | TATGTTTTAGAATAGTTACCACAACGTCGTGACTGGGAAA |
| pCN34_Rev_vra | TAAGTTTTTAATGACTTTCAAAAACGACGGCCAGTGAATT |
| vraUTSR_Fw | AATTCACTGGCCGTCGTTTTTGAAAGTCATTAAAAACTTAACAGG |
| vraUTSR_Rev | TTTCCCAGTCACGACGTTGTGGTAACTATTCTAAAACATATGGCA |
| For construction of PgraXRS complementation plasmid: | |
| pCN34_Fw_gra | AGAAGTTAGATAAAGAACATACAACGTCGTGACTGGGAAA |
| pCN34_Rev_gra | AAATGTACCACTCAATAACCAAAACGACGGCCAGTGAATT |
| graXRS_Fw | AATTCACTGGCCGTCGTTTTGGTTATTGAGTGGTACATTTGC |
| graXRS_Rev | TTTCCCAGTCACGACGTTGTATGTTCTTTATCTAACTTCTGTACC |
| For construction of PdltD complementation plasmid: | |
| pCN34_Fw_dlt | TATCTTTATAGGCGCCTTTGCGGAAAGAG |
| pCN34_Rev_dlt | AGCGCAAATAATTCGCCATTCAGGCTGC |
| dltA_pro_Fw | AATGGCGAATTATTTGCGCTGATGATAATTC |
| dltA_pro_Rev | TTAATTTCATATTAAGTCTCCCTCATTAGAAC |
| dltD_Fw | GAGACTTAATATGAAATTAAAACCTTTTTTACCC |
| dltD_Rev | CAAAGGCGCCTATAAAGATATTAAGTTAACAGAACATATTATG |
| For construction of Pcls2 complementation plasmid: | |
| pCN34_Fw_cls2 | GGCGCCTTTGCGGAAAGAG |
| pCN34_Rev_cls2 | ATTCGCCATTCAGGCTGC |
| cls2_Fw | AGCCTGAATGGCGAATCTGTTTAACGCCGAACGTG |
| cls2_Rev | TTCCGCAAAGGCGCCTTTCGTCAACGGTATCATGAAG |

values were divided by $OD_{600}$ measurements to correct for changes in cell density which occurred during the assay.

**WTA extraction and analysis.** TSB-grown and serum-adapted bacteria were generated as described above and WTA was extracted as described previously[100]. 40 ml cultures were washed with 40 ml 50 mM MES (pH 6.5) (Buffer 1) and resuspended in 30 ml 50 mM MES (pH 6.5) supplemented with 4% SDS (Buffer 2). Samples were boiled for 1 h and centrifuged before being washed twice in 2 ml Buffer 2, once in 2 ml 50 mM MES (pH 6.5) supplemented with 2% NaCl (Buffer 3), and once in 2 ml Buffer 1. The pellet was resuspended in 1 ml 20 mM Tris-HCl pH 8, 0.5% SDS and digested with 20 µg proteinase K for 4 h at 50 °C. The pellet was washed once with 1 ml Buffer 3, three times with 1 ml water, resuspended in 500 µl 0.1 M NaOH and incubated for 16 h at room temperature. After centrifugation, 500 µl supernatant was neutralised with 125 µl 1 M Tris-HCl (pH 7.8) and analysed by PAGE. 10 µl aliquots of WTA samples were separated on a 20% native polyacrylamide gel by electrophoresis using 0.1 M Tris, 0.1 M Tricine, pH 8.2 running buffer. Gels were then stained with alcian blue (1 mg/ml, 3% acetic acid), destained with water and imaged using a Gel Doc EZ Imager (Bio-Rad). The phosphate content of WTA extracts was determined as described previously[101]. 6 N $H_2SO_4$, water, 2.5% ammonium molybdate and 10% ascorbic acid were mixed in a ratio of 1:2:1:1. 100 µl of this was mixed with 100 µl WTA extracts and then incubated at 37 °C for 90 min. Phosphate concentrations were determined by measuring absorbance at 820 nm and correlating this to a standard curve generated from $NaH_2PO_4$ standards.

**Quantification of D-alanine concentrations in WTA extracts.** The amount of D-alanine modification present in WTA extracts was determined using a series of enzymatic reactions catalysed by D-amino acid oxidase and L-lactic dehydrogenase as described previously[102] with some modifications. Reactions were set up in a 96 well plate where each well contained 102 mM Tris-HCl (pH 8.5), 0.1875 U D-amino acid oxidase (Sigma), 20 U catalase (Sigma), 1.82 mg ml$^{-1}$ β-NADH, 2 U L-lactic acid dehydrogenase, and 20 µl WTA extract. Absorbance at 339 nm was measured before the addition of L-lactic acid dehydrogenase and then plates were incubated for 20 min at room temperature. $A_{339}$ was measured again and the value at 20 min subtracted from the value at 0 min. A standard curve was generated using known D-alanine concentrations (0–100 µM) and used to interpolate the D-alanine concentrations in WTA extracts.

**Extraction and purification of peptidoglycan.** Peptidoglycan was extracted from 300 ml cultures of TSB-grown or serum-adapted cultures of *S. aureus*. Pellets were recovered by centrifugation at $3200 \times g$ for 10 min, pooled, transferred to a 50 ml falcon tube and snap frozen in liquid nitrogen. 4% SDS (25 ml) was added to pellets and tubes were boiled in a beaker on a hot plate for 1 h. After cooling, pellets were recovered by centrifugation ($3200 \times g$ for 10 min), moved to 2 ml Eppendorf tubes and washed 6 times with 2 ml distilled water ($13,000 \times g$ for 3 min). After the final wash, pellets were resuspended in 2 ml 10 mM Tris HCl (pH 7.4) containing 2 mg ml$^{-1}$ pronase (Sigma). Tubes were incubated for 3 h at 60 °C to digest proteins. Samples were then resuspended in 30 ml 4% SDS and boiled for 30 min to inactivate the pronase. Pellets were washed 6 times with distilled water and freeze-dried. The freeze-dried cell walls were resuspended in 1 M HCl at 10 mg ml$^{-1}$. Samples were incubated at 37 °C for 5 h to remove WTA from the peptidoglycan by hydrolysis. After recovering the peptidoglycan by centrifugation ($13,000 \times g$ for 3 min), pellets were washed 6 times in distilled water, freeze dried and weighed.

**Rp-HPLC analysis of *S. aureus* peptidoglycan.** Five milligrams of peptidoglycan were resuspended in 400 µl of 20 mM phosphate buffer (pH 6.0) and digested in the presence of 125 µg of mutanolysin for 16 h at 37 °C. Mutanolysin was inactivated for 5 min at 100 °C and soluble disaccharide-peptides were recovered following centrifugation at $20,000 \times g$ for 10 min. After addition of one volume of 200 mM borate buffer (pH 9.25), muropeptides were reduced by the addition of 1 mg of sodium borohydride for 20 min at room temperature and the pH was adjusted to 4.5 with phosphoric acid.

For rp-HPLC analysis, the equivalent of 150 µg of soluble, reduced peptidoglycan fragments were injected on a Hypersil Gold aQ column (200 mm × 2.1 mm ID, 1.9 µm particles) equilibrated in 10 mM ammonium phosphate (pH 5.5) (buffer A) at a flow rate of 0.3 ml min$^{-1}$. Muropeptides were eluted with a 25 min gradient to 25% methanol in buffer A. Detection was carried out by monitoring absorbance at 210 nm.

**Triton X-100 induced lysis.** Cultures of TSB-grown and serum-adapted *S. aureus* were generated as described above, washed twice in PBS and resuspended in PBS supplemented with 0.05% Triton X-100. 200 µl aliquots (at $10^8$ CFU ml$^{-1}$) were transferred to flat-bottomed 96-well plates, incubated at 30 °C and $OD_{600}$ measured every 15 min for 6 h. Values were blanked against uninoculated wells and calculated as a percentage of the starting value.

**Analysis of cell wall hydrolase activity by zymography**. 10 ml cultures of TSB-grown or serum-adapted *S. aureus* were generated as described above, resuspended in 1 ml 50 mM Tris-HCl (pH 7.5) containing 20% sucrose, 10 mM $MgCl_2$ and 0.1 µg ml$^{-1}$ lysostaphin and incubated at 37 °C for 1 h. Samples were centrifuged ($8000 \times g$, 3 min) and supernatants (containing cell wall fragments) were analysed by SDS-PAGE using 10% polyacrylamide gels containing autoclaved *S. aureus*. Gels were washed in water, incubated overnight at 37 °C in renaturation buffer (50 mM Tris-HCl (pH 7.5), 0.1% Triton X-100, 10 mM $CaCl_2$, 10 mM $MgCl_2$), stained with 0.1% methylene blue in 0.01% KOH for 1 h at room temperature and destained overnight with water. For an example of presentation of full gel images, see the Source Data file.

**Lipid extraction and thin layer chromatography**. Lipids were extracted from TSB-grown (10 ml) and serum-adapted (10 ml) cultures using a method based on that of Bligh and Dyer[59,60]. Cells were washed twice with PBS, twice with 0.1% Triton X-100 and twice with PBS to remove unincorporated serum-derived lipids[35] before resuspension in 300 µl 2% NaCl. After addition of 0.2 mg ml$^{-1}$ lysostaphin, cells were incubated at 37 °C for 10 min. 1 ml chloroform-methanol (2:1; v/v) was added, vortexed for 2 min and incubated at room temperature for 30 min. Chloroform (300 µl) and 2% NaCl (300 µl) were added, and after centrifugation, the lower layer was recovered and evaporated at room temperature. Samples were resuspended in 100 µl chloroform-methanol (2:1; v/v) and separated by one-dimensional TLC. Equal volumes of lipid extracts were spotted onto silica 60 F254 HPTLC plates (Merck) and migrated with chloroform:methanol:ammonium hydroxide (30%) (65:30:4; v/v/v). After drying, TLC plates were sprayed with $CuSO_4$ (100 mg ml$^{-1}$) in 8% phosphoric acid and heated at 180 °C. Signal intensities were quantified using ImageJ.

**Construction of *S. aureus* complementation strains**. *S. aureus* mutants were complemented using the multi-copy *E. coli*/*S. aureus* shuttle vector pCN34[103] containing the relevant gene/operon along with its promoter and terminator regions. All genes were amplified from USA300 LAC* genomic DNA and were inserted into pCN34 at the multiple cloning site using Gibson assembly. Primer sequences used to generate complementation strains are shown in Table 2.

The *vraUTSR* operon, together with 500 bp regions up and downstream of the operon, was amplified using primers *vraUTSR*_Fw and *vraUTSR*_Rev. The pCN34 vector was amplified using primers pCN34_Fw_*vra* and pCN34_Rev_*vra*. The *graXRS* operon, together with 500 bp regions up and downstream of the operon, was amplified using primers *graXRS*_Fw and *graXRS*_Rev. The pCN34 vector was amplified using primers pCN34_Fw_*gra* and pCN34_Rev_*gra*. The *dltD* gene and the *dlt* operon promoter (present in the 500 bp region upstream of *dltA*) were amplified separately. *dltD*, together with a 500 bp region downstream of *dltD*, was amplified using primers *dltD*_Fw and *dltD*_Rev. The 500 bp region upstream of *dltA* was amplified using primers *dlt*_pro_Fw and *dlt*_pro_Rev. The pCN34 vector was amplified using primers pCN34_Fw_*dlt* and pCN34_Rev_*dlt*. The *cls2* gene, along with upstream and downstream regions, was amplified using primers *cls2*_Fw and *cls2*_Rev. The pCN34 vector was amplified using primers pCN34_Fw_*cls2* and pCN34_Rev_*cls2*.

After amplification, fragments were assembled using Gibson assembly, transformed into the *E. coli* strain DH5α, electroporated into the restriction-deficient *S. aureus* strain RN4220 and transduced (along with empty pCN34 as a control) by phage φ11 into the relevant mutant.

**Statistical analyses**. CFU data were log$_{10}$ transformed and presented as the geometric mean ± geometric standard deviations. Non-CFU data were presented as the mean ± standard deviation. All experiments consisted of three or more independent replicates and were analysed by Student's *t*-test, one-way ANOVA or two-way ANOVA with appropriate *post-hoc* multiple comparison test using GraphPad Prism (V8.0), as described in the figure legends. P values calculated to 4 decimal places.

**Reporting summary**. Further information on research design is available in the Nature Research Reporting Summary linked to this article.

## Data availability

Source data are provided with this paper.

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

## Acknowledgements

Angelika Grundling and Nathalie Reichmann (Imperial College London) are thanked for providing strains. Vladimir Pelicic (Imperial College London) is thanked for providing access to the fluorescent microscope. Simon Foster (University of Sheffield) is thanked for providing HADA. EVKL was supported by a Wellcome Trust PhD Studentship (203812/Z/16/Z). AME acknowledges funding from the Rosetrees Trust and from the Imperial NIHR Biomedical Research Centre, Imperial College London. All authors acknowledge the provision of strains by the Network on Antimicrobial Resistance in Staphylococcus aureus (NARSA) Program: under NIAID/ NIH Contract No. HHSN272200700055C. The funders had no role in the study design, interpretation of the findings or the writing of the manuscript.

## Author contributions

E.V.K.L., S.M., and A.M.E. conceived experiments, analysed data and wrote the manuscript. E.V.K.L. and S.M. performed experiments.

## Competing interests

The authors declare no competing interests.
