## [Peer Review File · Nature Communications]

Reviewers' Comments:

Reviewer #1:

Remarks to the Author:

The manuscript by Ledger et al. describes the interesting phenomenon whereby serum is inducing daptomycin tolerance in *S. aureus*. The research area is interesting and the role of serum in antibiotic activities is potentially very important. The paper is also very well-written and if the authors are correct, the findings are very important.

However, I have some major reservations about this study, particularly regarding the comparison of TSB exponential phase bacteria with serum-adapted bacteria, which is the principal assay that essentially all the findings in the paper are reliant upon. The serum adaptation takes place over 16 hours before the bacteria are exposed to antibiotic and this is then compared to bacteria grown in TSB to exponential phase. This 16 hour incubation will render the bacteria starved and in a stationary phase of growth. A more relevant comparison would have been to leave the bacteria in TSB for the additional 16 hours and then compare that with 16h serum exposed bacteria. It's very possible that many of the effects observed by the authors are stationary phase specific and not serum specific. This is relevant to every figure in the paper. If the authors can re-do the assay in a more like-for-like way and show that indeed the serum is the essential component and that incubation in TSB and PBS for this length of time would not lead to the same tolerance and then remove the tolerance inducing effect by denaturing or fractionating the serum, this would convince me of the veracity of their claims and make the study important and robust.

In addition to this critical concern, I have concerns regarding Fig.3 specifically, as follows:

Many graphs are lacking appropriate statistical analysis. Also, it would be helpful to see the control cfu raw data for figures. Percentages can often be misleading. Are there differences in overall survival in serum in the absence of drug for various mutants in Fig. 3A for instance?

I am curious of the data presented in figure 3G. This is definitely a scenario where the raw cfu are essential and the statistics ought to be performed on the raw cfu values. Although the authors declare a dose dependent impact of the antibody, the average survival they show at 0 antibody is remarkably high in this iteration of the experiment, averaging over 20%. At the highest concentration of antibody, it averages at 5% and this decrease is apparently significant. In other experiments however, the WT in serum survived 3% (Fig. 3A) and 5% (Fig 3D). Therefore, the highest antibody concentration is still resulting in the usual tolerance seen in serum in other experiments. Could the author explain the inconsistencies between these apparently identical experiments?

Reviewer #2:

Remarks to the Author:

The manuscript by Ledger et al. is overall a really interesting story and a pleasant read. It is apparent that the authors have made a considerable effort to thoroughly test and control their hypotheses and they have succeeded in providing a solid line of argumentation based on a sound methodology. While the suitability for the journal remains to be judged by the editor, it is my opinion that this work presents a significant contribution to the field.

However, I have a number of comments that should be addressed before the paper is considered for publication.

1. One of my major issues with this paper is that it refers to outdated data for the mechanism of daptomycin, which actually has implications on the interpretation of the results (I will come back to that). The authors fail to acknowledge that daptomycin does not only target PG but also lipid-linked cell wall precursors and affects the localization of cell wall synthesis enzymes like MurG (PMIDs 32193379, 27791134). This should be acknowledged and discussed. Further, the authors refer to the permeabilizing effect of daptomycin, yet this effect only occurs well above minimal inhibitory concentration range (reviewed in PMID 31947747) and is rather a consequence of cell death than crucial for its mechanism (at least under lab conditions). I would wish that the authors

acknowledge these newer models of daptomycin action and, importantly, discuss the implications of its cell wall-inhibitory effects for their study.

2. Related to that, is the observed tolerance only due to limitation of target access (by cell wall thickening and increased CL) or could there maybe also be an increase in lipid II and other bactoprenol-linked cell wall precursors targeted by daptomycin that could play a role here? This could, for example, be an alternative explanation for the authors' findings on fosfomycin and this possibility should be discussed.

3. Also related to that, at least in terms of concentration-dependent effect of daptomycin, I think the authors should be a bit careful with their interpretations regarding the mechanism of daptomycin. The authors use 80 µg/ml daptomycin, which is explained and justified and thus per se fine, but one needs to keep in mind that this is a huge overkill for cells grown under lab conditions (I guess probably about 80x MIC in TSB) while it is barely inhibitory for the serum-adapted cells. This should be made clearer in the text and considered in the interpretation of the results. For example, after 6 h at this concentration, a log₆ reduction in CFU is observed. Then it is no wonder that under the same conditions, the authors observed membrane depolarization and PI uptake, since dead cells will certainly be positive for both assays (and daptomycin does not permit PI uptake at around MIC, only at suprainhibitory concentrations, see references above). The way these assays were performed, they rather support the non-susceptibility of serum-adapted cells than give insight into the mechanism. Please acknowledge this point (it does not take away much from the observation that these effects are not observed in the serum-adapted cultures).

4. Also related to concentrations, 80 µg/ml was chosen based on serum concentrations of daptomycin, but how was the CFU count chosen? Does it reflect the bacterial load during infection? Similarly, are LL-37 concentrations chosen here representative of what bacteria would encounter during infection?

5. Have the authors considered the impact of calcium concentrations in the different media? I assume that it was insured that enough calcium was present for optimal daptomycin activity, but too much calcium can damage bacterial cell membranes. Do the authors know the exact calcium concentrations under each condition and are they comparable?

6. It has been shown that GraRS plays a role for virulence in *S. aureus* (PMID 34484169). The authors may want to discuss this in the context of their findings.

7. Out of interest, can the authors elaborate a bit more on their findings regarding the other compound affected by serum adaptation? Dap, Nis, Van, and Gra all affect the cell membrane and/or cell wall, but Nit and Gen have intracellular targets and no major effects on the cell envelope. Is it just the thicker cell walls then that cause this effect? Or are they also affected by CL content? Is their activity affected by a GraRS deletion? Do they maybe induce GraRS?

Minor comments:

8. While the paper is generally well-written, it should be checked again with regard to the use of commas and hyphenation.

9. Line 96: This statement seems to be conflicting with the previous statement about persisters.

10. Line 132: Can the authors explain why they chose 16 h?

11. Fig 2B: The image quality is not great. The cells are out of focus and the images appear pixelated.

12. Line 344-346: Can the authors further discuss this point?

13. Line 609: Following the latest models, daptomycin can be classified as cell wall synthesis inhibitor as well. Maybe this should be phrased differently to accommodate the newer findings.

Reviewers' comments in black text

Our responses in blue, with related changes highlighted in yellow in the manuscript.

Text highlighted in green in the manuscript was edited to conform with the journal's editorial guidelines.

REVIEWER COMMENTS

Reviewer #1 (Remarks to the Author):

The manuscript by Ledger et al. describes the interesting phenomenon whereby serum is inducing daptomycin tolerance in *S. aureus*. The research area is interesting and the role of serum in antibiotic activities is potentially very important. The paper is also very well-written and if the authors are correct, the findings are very important.

We thank the reviewer for their time in evaluating our manuscript and for their very positive comments.

However, I have some major reservations about this study, particularly regarding the comparison of TSB exponential phase bacteria with serum-adapted bacteria, which is the principal assay that essentially all the findings in the paper are reliant upon. The serum adaptation takes place over 16 hours before the bacteria are exposed to antibiotic and this is then compared to bacteria grown in TSB to exponential phase. This 16 hour incubation will render the bacteria starved and in a stationary phase of growth. A more relevant comparison would have been to leave the bacteria in TSB for the additional 16 hours and then compare that with 16h serum exposed bacteria. It's very possible that many of the effects observed by the authors are stationary phase specific and not serum specific. This is relevant to every figure in the paper.

If the authors can re-do the assay in a more like-for-like way and show that indeed the serum is the essential component and that incubation in TSB and PBS for this length of time would not lead to the same tolerance and then remove the tolerance inducing effect by denaturing or fractionating the serum, this would convince me of the veracity of their claims and make the study important and robust.

We have followed the reviewer's suggestion and assessed daptomycin tolerance in bacteria incubated in serum, TSB or PBS for 16 h. The data show that bacteria incubated in TSB or PBS for 16 h have a significantly lower level of daptomycin tolerance than bacteria incubated in serum for the same length of time (please see supplementary figure 2 and lines: 150-155).

These findings agree with data present in the original manuscript showing that incubation of *S. aureus* in RPMI 1640 for 16 hours did not trigger daptomycin tolerance unless it was supplemented with LL-37 or colistin (Fig. 3f, supplementary figure 6).

Furthermore, these data are in keeping with work from Mascio *et al.*, 2007 (AAC, PMID: 17923487), who showed that daptomycin retains a high degree of bactericidal activity against stationary phase *S. aureus*

Therefore, the high levels of daptomycin tolerance observed in serum-adapted *S. aureus* is not simply a consequence of starvation or entering stationary phase.

We have also followed the reviewer's suggestion to fractionate the serum to further confirm the role of a specific serum component in triggering tolerance (please see Fig. S8 and lines: 278-285). To do

this, serum was dialysed against PBS using a range of molecular weight cut-off filters to remove various sized molecules, and then tested for its ability to trigger daptomycin tolerance.

All fractions arrested *S. aureus* growth. However, daptomycin tolerance was lost when bacteria were incubated in serum dialysed against a 5 kDa cut off filter, indicative of a low MW trigger. By contrast, serum that had been dialysed against a 0.5 kDa cut off filter retained the ability to trigger tolerance, indicating a serum component >0.5 kDa and <5 kDa was a specific trigger.

To further explore this, we incubated *S. aureus* for 16 hours in water alone, or water containing serum fractions of < 5 kDa, or >5 kDa. This demonstrated that the <5 kDa fraction is necessary and sufficient to trigger daptomycin tolerance in *S. aureus* in water. By contrast, incubation of *S. aureus* in water alone or water containing the >5 kDa serum fraction did not trigger tolerance.

Taken together, the experiments in Supplementary figure 8 indicated that a serum component in the 0.5–5 kDa mass range was responsible for triggering tolerance. This agrees with our other data showing that LL-37 (4.5 kDa) is the serum component that triggers tolerance (please see supplementary figure 8 and lines: 278-285).

These serum fractionation studies also provide additional evidence that serum-triggered induction of daptomycin tolerance is not due to a starvation response since incubation of *S. aureus* in PBS or water for 16 h did not trigger daptomycin tolerance (supplementary figure 8).

In addressing the reviewer's question of whether serum specifically triggers daptomycin tolerance, we would also like to draw attention to data in Fig. 3g, which shows that the ability of serum to trigger tolerance is compromised by the presence of an anti-LL-37 antibody. By contrast, an identical concentration of an anti-hNP-1 antibody had no effect on tolerance. If serum-induced daptomycin tolerance was simply a starvation/stationary response this would be expected to occur in the presence of either antibody.

Further evidence for the specificity of serum in triggering tolerance comes from experiments using a *graS::Tn* mutant and *PdltA-gfp* reporter. In these, we show that daptomycin tolerance requires activation of the GraRS two-component system, which occurs in serum but not RPMI 1640, unless this medium is supplemented with LL-37 or colistin (an established trigger of GraRS) (Fig. 3c,d,e,f & Fig. S6).

Together, these new data complement our previous work by providing additional evidence that serum specifically induces daptomycin tolerance, via LL-37 mediated activation of GraRS and subsequent cell wall accumulation and via cardiolipin accumulation catalysed by Cls2. These findings are supported by work from Hines *et al.*, 2020 (*mSphere*, PMID: 32554713) who reported that serum supplementation of TSB led to cell wall and cardiolipin accumulation. Whilst these studies were done with mid-exponential phase cultures and did not assess the mechanisms involved or antibiotic tolerance, they do provide additional evidence that serum can specifically trigger changes similar to those we demonstrate are responsible for serum-induced daptomycin tolerance.

In addition to this critical concern, I have concerns regarding Fig.3 specifically, as follows: Many graphs are lacking appropriate statistical analysis. Also, it would be helpful to see the control cfu raw data for figures. Percentages can often be misleading. Are there differences in overall survival in serum in the absence of drug for various mutants in Fig. 3A for instance?

We thank the reviewer for raising this point. We have carefully reviewed our statistical analyses of the data, drawing on guidance provided in Olsen, 2003 Immunity and Infection (PMID: 14638751) which describes approaches suitable for analysis of CFU counts such as ours.

As requested, we have revised all figures that showed % survival to now show CFU counts, which have been log transformed to make them compatible with the parametric ANOVA analyses used to analyse the data. The raw, untransformed CFU counts are supplied as supplementary data. We have repeated statistical analyses for all graphs that show CFU counts and highlighted significant differences. The figure legends have been revised to ensure all necessary information on statistical tests is included.

These revised analyses strengthen our conclusions that serum specifically induces daptomycin tolerance via GraRS activation and accumulation of cardiolipin.

To address the third point about Fig. 3a, we have included data showing that the two mutants with reduced daptomycin tolerance (*vraRS* & *graRS*) do not have a survival deficit in serum in the absence of antibiotic (please see Fig. S4 and S5).

I am curious of the data presented in figure 3G. This is definitely a scenario where the raw cfu are essential and the statistics ought to be performed on the raw cfu values. Although the authors declare a dose dependent impact of the antibody, the average survival they show at 0 antibody is remarkably high in this iteration of the experiment, averaging over 20%. At the highest concentration of antibody, it averages at 5% and this decrease is apparently significant. In other experiments however, the WT in serum survived 3% (Fig. 3A) and 5% (Fig 3D). Therefore, the highest antibody concentration is still resulting in the usual tolerance seen in serum in other experiments. Could the author explain the inconsistencies between these apparently identical experiments?

As described above, we have revised all relevant graphs to show log transformed CFU numbers instead of % survival and repeated our statistical analyses on these data. This revised analysis confirms that the anti-LL-37 antibody causes a significant, dose-dependent reduction in serum-induced tolerance, in contrast to identical concentrations of an anti-hNP-1 antibody that did not affect serum-induced tolerance (Fig. 3g).

As shown in the graph below, we did see some assay-to-assay variation in the degree to which serum triggered tolerance, most likely due to batch-to-batch variation between the sera used. We also observed some variation in the final CFU counts of bacteria grown in TSB, similar to what has been reported previously (e.g. Mechler *et al.*, 2016 AAC PMID: 26883712). However, in all cases, serum triggered tolerance to a very high degree and bacterial survival was always greater than that seen in TSB-grown cells.

To address this variation, each individual experiment used a single batch of serum to enable fair comparisons to be made e.g. between wild type and mutant or different treatment groups. A note to this effect is now included in the revised manuscript (please see lines: 712-713).

In the specific example of Fig. 3g, we undertook a direct comparison between the same batch of serum pre-treated with an anti-LL-37 antibody or an anti-hNP-1 antibody. Whilst the anti-LL-37 antibody caused a significant and dose-dependent inhibition of tolerance induction, the anti-hNP-1 antibody had no effect on tolerance. This demonstrates a specific role for LL-37 in triggering daptomycin tolerance and is in keeping with other experiments showing that LL-37 (but not other human AMPs) triggered tolerance in *S. aureus* cells incubated in RPMI 1640 (Fig. S9).

With reference to the average survival of serum-adapted cells in Fig. 3g, this falls within the range of values described in the manuscript and is not unusually high. As shown below, the range of values for survival of daptomycin-exposed serum-adapted *S. aureus* obtained throughout the manuscript is 1.3 % - 58 % with a median of 12 %. By contrast, survival of TSB-grown cells was 0.000045 % - 0.004% with a median of 0.0002 %.

Graph shows the % survival of *S. aureus* after 6 h exposure to daptomycin having been previously grown in TSB or incubated in serum. Each dot relates to the mean average of values from a single experiment contained in the manuscript.

Reviewer #2 (Remarks to the Author):

The manuscript by Ledger et al. is overall a really interesting story and a pleasant read. It is apparent that the authors have made a considerable effort to thoroughly test and control their hypotheses and they have succeeded in providing a solid line of argumentation based on a sound methodology. While the suitability for the journal remains to be judged by the editor, it is my opinion that this work presents a significant contribution to the field.

We thank the reviewer for their time in evaluating our manuscript and for their very positive comments.

However, I have a number of comments that should be addressed before the paper is considered for publication.

1. One of my major issues with this paper is that it refers to outdated data for the mechanism of daptomycin, which actually has implications on the interpretation of the results (I will come back to that). The authors fail to acknowledge that daptomycin does not only target PG but also lipid-linked cell wall precursors and affects the localization of cell wall synthesis enzymes like MurG (PMIDs 32193379, 27791134). This should be acknowledged and discussed. Further, the authors refer to the permeabilizing effect of daptomycin, yet this effect only occurs well above minimal inhibitory concentration range (reviewed in PMID 31947747) and is rather a consequence of cell death than crucial for its mechanism (at least under lab conditions). I would wish that the authors acknowledge these newer models of daptomycin action and, importantly, discuss the implications of its cell wall-inhibitory effects for their study.

We have revised the introduction to make clear that, in addition to phosphatidylglycerol, daptomycin also targets cell wall precursors and affects the localisation of enzymes involved in cell wall synthesis. We also make clear that membrane disruption occurs at high concentrations of antibiotics and is not required for anti-bacterial activity (please see lines: 56-61).

We have also expanded the discussion section to better reflect the fact that daptomycin disrupts cell wall biosynthesis (please see lines: 581-585).

These changes do not alter the key conclusions of the manuscript, that serum induces accumulation of cell wall and cardiolipin that prevent daptomycin binding to its membrane targets and results in high levels of daptomycin tolerance.

2. Related to that, is the observed tolerance only due to limitation of target access (by cell wall thickening and increased CL) or could there maybe also be an increase in lipid II and other bactoprenol-linked cell wall precursors targeted by daptomycin that could play a role here? This could, for example, be an alternative explanation for the authors' findings on fosfomycin and this possibility should be discussed.

We believe that tolerance is due to cell wall thickening and increased membrane cardiolipin content, which prevents daptomycin from binding its membrane targets of phosphatidylglycerol and lipid II. Inhibition of both cell wall and cardiolipin accumulation reduces the tolerance of serum-adapted *S. aureus* to levels seen for TSB-grown cells, suggesting that they fully explain the tolerance phenotype.

We don't believe that increased lipid II abundance would confer tolerance since the Grein *et al.*, 2020 *Nat Comm* (PMID 32193379) paper mentioned above demonstrates that increases in lipid II abundance sensitise *S. aureus* to daptomycin. By contrast, a reduction in free lipid II (via sequestration with teicoplanin) promoted daptomycin tolerance.

Since we would expect fosfomycin to reduce the abundance of lipid II (PMID: 26620564), which in turn would be expected to increase daptomycin tolerance, we do not think this is an explanation for synergy.

However, it is possible that lipid II may become depleted during *S. aureus* incubation in serum as we observed a large increase in both the WTA and peptidoglycan contents of serum-adapted bacteria, and this may contribute to the reduced daptomycin binding and susceptibility that we observed. This possibility is now discussed in lines 582-583.

3. Also related to that, at least in terms of concentration-dependent effect of daptomycin, I think the authors should be a bit careful with their interpretations regarding the mechanism of daptomycin. The authors use 80 µg/ml daptomycin, which is explained and justified and thus per se fine, but one needs to keep in mind that this is a huge overkill for cells grown under lab conditions (I guess probably about 80x MIC in TSB) while it is barely inhibitory for the serum-adapted cells. This should be made clearer in the text and considered in the interpretation of the results. For example, after 6 h at this concentration, a log₆ reduction in CFU is observed. Then it is no wonder that under the same conditions, the authors observed membrane depolarization and PI uptake, since dead cells will certainly be positive for both assays (and daptomycin does not permit PI uptake at around MIC, only at suprainhibitory concentrations, see references above). The way these assays were performed, they rather support the non-susceptibility of serum-adapted cells than give insight into the mechanism. Please acknowledge this point (it does not take away much from the observation that these effects are not observed in the serum-adapted cultures).

We have edited the text to make clear that the concentrations of antibiotic used, whilst clinically relevant are well above the MIC (please see lines: 198-199). However, we believe the fact that we see tolerance at such a high concentration of daptomycin underscores the strength of the tolerance phenotype.

The assays measuring membrane depolarisation and permeabilization were done to provide additional evidence that daptomycin was unable to access the membrane, rather than trying to gain insight into the mechanism of action of the antibiotic. This is now made clear in the manuscript (please see lines: 198-200 and removal of line referring to membrane damage as causative of bacterial death).

4. Also related to concentrations, 80 µg/ml was chosen based on serum concentrations of daptomycin, but how was the CFU count chosen? Does it reflect the bacterial load during infection? Similarly, are LL-37 concentrations chosen here representative of what bacteria would encounter during infection?

Since daptomycin is licenced to treat endocarditis, a common and hard to treat manifestation of *S. aureus* bacteraemia, we used an inoculum of 2×10^8 CFU ml⁻¹, which is a reasonable approximation the concentration of bacteria in a gram of tissue. It was also important to use a high concentration of bacteria to provide a large dynamic range to enable the detection of tolerance and subsequently dissect the mechanisms responsible. We have now made this clear in the revised manuscript (please see lines: 131-134).

We have included a discussion of the concentration of LL-37 used in these studies and its physiological relevance (please see lines: 605-611).

5. Have the authors considered the impact of calcium concentrations in the different media? I assume that it was insured that enough calcium was present for optimal daptomycin activity, but too much calcium can damage bacterial cell membranes. Do the authors know the exact calcium concentrations under each condition and are they comparable?

In keeping with previous work (e.g. Grein *et al.*, 2020, PMID 32193379), we used the standard concentration of 50 µg ml⁻¹ Ca²⁺ in laboratory media, which is required for anti-bacterial activity and is physiologically relevant, since the free Ca²⁺ concentration of serum is 44 – 56 µg ml⁻¹. This concentration is sufficient for full activity of the antibiotic, and this is now made clear in the revised manuscript (please see lines: 704-706).

Since bacteria survived well in serum (Fig. S1, S4) and had a very low signal from propidium iodide in the absence of daptomycin (Fig. 2d) we do not believe there was significant membrane damage in the absence of antibiotics.

6. It has been shown that GraRS plays a role for virulence in *S. aureus* (PMID 34484169). The authors may want to discuss this in the context of their findings.

The original manuscript contained a discussion of this important point (please see lines 612-614), which we have expanded upon (please see lines: 595-596).

7. Out of interest, can the authors elaborate a bit more on their findings regarding the other compound affected by serum adaptation? Dap, Nis, Van, and Gra all affect the cell membrane and/or cell wall, but Nit and Gen have intracellular targets and no major effects on the cell envelope. Is it just the thicker cell walls then that cause this effect? Or are they also affected by CL content? Is their activity affected by a GraRS deletion? Do they maybe induce GraRS?

We have expanded the discussion section to consider potential mechanisms underpinning tolerance to these antimicrobials in serum adapted *S. aureus* (please see lines: 655-663).

Minor comments:

8. While the paper is generally well-written, it should be checked again with regard to the use of commas and hyphenation.

We have carefully reviewed the manuscript and corrected it where necessary (please see highlighted text in marked up copy of the manuscript).

9. Line 96: This statement seems to be conflicting with the previous statement about persisters.

It's not clear what is meant here since there isn't a mention of persisters until line 100 (line 103 in the revised manuscript).

10. Line 132: Can the authors explain why they chose 16 h?

We chose this as a reasonable estimate for the time from infection onset to treatment. We allowed 4 hours for symptom onset and 12 hours for diagnosis by blood culture [Ruiz-Giardin et al., IJID 2015]. This is now explained in the methods section (please see lines: 711-713).

11. Fig 2B: The image quality is not great. The cells are out of focus and the images appear pixelated.

We have increased the resolution of the figure.

12. Line 344-346: Can the authors further discuss this point?

We have expanded the discussion section to cover this in more detail (please see lines: 624-629).

13. Line 609: Following the latest models, daptomycin can be classified as cell wall synthesis inhibitor as well. Maybe this should be phrased differently to accommodate the newer findings.

We have edited the text accordingly (please see line: 677).

Reviewers' Comments:

Reviewer #1:

Remarks to the Author:

The reviewers have carefully and comprehensively addressed my concerns and the manuscript is improved and of high quality.

Reviewer #2:

Remarks to the Author:

The authors have delivered a carefully revised manuscript, which addresses all my concerns appropriately. I have no further comments.

Response to reviewers' comments

Reviewers' comments in black text

Our responses in blue

Reviewer #1 (Remarks to the Author):

The reviewers have carefully and comprehensively addressed my concerns and the manuscript is improved and of high quality.

Reviewer #2 (Remarks to the Author):

The authors have delivered a carefully revised manuscript, which addresses all my concerns appropriately. I have no further comments.

We thank the reviewers for their time and effort in evaluating our manuscript. We are delighted that they are satisfied that our revisions have fully addressed their concerns.